# FGFR1 Inhibition by Pemigatinib Enhances Radiosensitivity in Glioblastoma Stem Cells Through S100A4 Downregulation

**DOI:** 10.3390/cells14181427

**Published:** 2025-09-11

**Authors:** Valérie Gouazé-Andersson, Caroline Delmas, Yvan Nicaise, Julien Nicolau, Juan Pablo Cerapio, Elizabeth Cohen-Jonathan Moyal

**Affiliations:** 1Oncopole Claudius Regaud, IUCT-Oncopole Toulouse, 31037 Toulouse, France; delmas.caroline@iuct-oncopole.fr (C.D.); moyal.elizabeth@iuct-oncopole.fr (E.C.-J.M.); 2CNRS, Inserm, CRCT, University Toulouse, 31037 Toulouse, France; yvan.nicaise@inserm.fr (Y.N.); juan-pablo.cerapio-arroyo@inserm.fr (J.P.C.); 3Neurosurgery Department, Toulouse University Hospital, 31059 Toulouse, France; nicolau.j@chu-toulouse.fr; 4Faculty of Medicine, Paul Sabatier University—Toulouse III, 31400 Toulouse, France

**Keywords:** glioblastoma, cancer stem cells, FGFR1 inhibition, pemigatinib, radioresistance, S100A4, tumor resistance, targeted therapy

## Abstract

Glioblastoma (GBM) is an aggressive and highly heterogeneous tumor that frequently recurs despite surgery followed by radio-chemotherapy and, more recently, TTFields. This recurrence is largely driven by glioblastoma stem cells (GSCs), which are intrinsically resistant to standard therapies. Identifying molecular targets that underlie this resistance is therefore critical. Here, we investigated whether the inhibition of FGFR1, previously identified as a key mediator of GBM radioresistance, using pemigatinib, a selective FGFR1–3 inhibitor, could enhance GSC radiosensitivity in vitro and in vivo. Pemigatinib treatment inhibited FGFR1 signaling, promoted proteasome-dependent FGFR1 degradation, and reduced the viability, neurosphere formation, and sphere size in GSCs with unmethylated MGMT, a subgroup known for poor response to standard treatments. In MGMT-unmethylated differentiated GBM cell lines, pemigatinib combined with temozolomide further enhanced radiosensitivity. Transcriptomic analysis revealed that pemigatinib treatment led to the downregulation of *S100A4,* a biomarker associated with mesenchymal transition, angiogenesis, and immune modulation in GBM. Functional studies confirmed that silencing *S100A4* significantly improved GSCs’ response to irradiation. In vivo, pemigatinib combined with localized irradiation produced the longest median survival compared to either treatment alone in mice bearing orthotopic GSC-derived tumors, although the difference was not statistically significant. These findings support further clinical investigation to validate these preclinical findings and determine the potential role of FGFR inhibition as part of multimodal GBM therapy.

## 1. Introduction

Glioblastoma multiforme (GBM) is the most prevalent and deadly primary brain tumor in adults. It is classified into three molecular subtypes: proneural, classical (CL), and mesenchymal (MES) [1]. The MES subtype is the most aggressive, while CL has an intermediate prognosis. Standard treatment includes surgical resection, radiotherapy, and temozolomide (TMZ) [2]. Since 2017, the addition of Tumor Treating Fields (TTFields), which deliver alternating electrical fields to the skull, has improved progression-free and overall survival [3,4], independent of MGMT methylation status [5]. Still, median survival remains about 21 months, largely due to tumor heterogeneity and resistance to radiotherapy [4]. A major contributor to treatment failure is the presence of GBM stem cells (GSCs), self-renewing, highly tumorigenic cells with strong radioresistance, linked to enhanced DNA damage repair and plasticity [6,7,8]. Our prior studies have highlighted the critical role of FGFR1 in mediating radioresistance in GBM. In a clinical trial combining tipifarnib with radiotherapy, we showed that high FGFR1 expression correlated with worse overall and progression-free survival [9]. In vitro studies have demonstrated that targeting FGFR1 reduces GBM radioresistance. In differentiated GBM cells, combining FGFR1 inhibition with radiation has been shown to introduce centrosome overduplication, mitotic catastrophe, and a decrease in HIF-1α levels, thereby enhancing radiosensitivity [10]. While HIF-1α is a key mediator of hypoxic adaptation, it also independently promotes radioresistance through the transcriptional regulation of survival pathways [11,12]. In vivo, FGFR1 inhibition delays the growth of irradiated tumor xenografts, a process linked to reduced HIF-1α levels, without affecting blood vessel integrity [10]. In GSCs, FGFR1 is also crucial in radioresistance, with its inhibition significantly enhancing radiosensitivity. This radiosensitization is associated with downregulated expression of FOXM1, a gene involved in both chemoresistance and radioresistance. The FGFR1/FOXM1 pathway is also implicated in the mesenchymal transition, as silencing *FGFR1* or *FOXM1* in GSCs downregulates genes like *GLI2*, *TWIST1*, and *ZEB1*, which are involved in this process. This inhibition also significantly reduces GSC migration, indicating that the FGFR1/FOXM1 pathway, which governs GSC radioresistance, is also a key driver of mesenchymal transition. Taken together, these results, along with previously reported data in differentiated cells, clearly established that the FGFR1-FOXM1-dependent pathway in GSCs plays a pivotal role in GBM treatment resistance [13]. In clinical samples comparing primary and recurrent GBM, varying expression levels of targets such as ALK, PDGFRβ, PDGFRA, MET, FGFR1, FGFR2, and FGFR3 have been observed, indicating significant heterogeneity throughout the disease’s progression. Notably, FGFR1 is the only target that maintains consistent expression in tissue samples from GBM patients who have undergone radiotherapy and TMZ treatment [14]. Recent cluster interaction and spatial transcriptomic analyses have uncovered a critical role for stem-like glioblastoma cells in shaping the tumor microenvironment. These cells secrete specific chemokines that recruit early myeloid-derived suppressor cells (E-MDSCs), which in turn release growth factors that can fuel tumor progression. Among the signaling pathways implicated, the FGF11–FGFR1 axis has emerged as a potential key driver of glioblastoma growth and resistance [15]. Furthermore, an interaction between FGFR1 and integrin α6 has been identified, demonstrating that α6-integrin contributes to GSC proliferation and stemness by regulating FGFR1 and FOXM1 expression via the ZEB1/YAP1 transcriptional complex [16]. This α6-integrin–FGFR1 crosstalk underscores the importance of the FGFR1 axis in sustaining glioblastoma stem cell properties and supports its relevance as a therapeutic target in GBM. Pemigatinib, also known as INCB054828, distinguishes itself from earlier FGFR inhibitors through its high potency and selectivity for FGFR1–3, with reported IC_50_ values of approximately 0.4 nM for FGFR1, 0.5 nM for FGFR2, and 1.0 nM for FGFR3, while showing markedly weaker activity against FGFR4 (IC_50_ ≈ 30 nM). In a broad kinase selectivity screen including more than 100 kinases, pemigatinib exhibited over 100-fold selectivity for FGFR1–3 relative to the vast majority of other targets, with only a few non-FGFR kinases with IC_50_ values below 1000 nM, including VEGFR-2 (KDR; IC_50_ ≈ 190 nM) and c-Kit (IC_50_ ≈ 270 nM) [17]. These values highlight its ultra-low nanomolar biochemical potency and strong target specificity. In our study, however, higher concentrations were required to elicit functional effects in glioblastoma stem cells, likely reflecting the intrinsic resistance mechanisms and reduced drug permeability characteristic of GSCs compared to conventional cancer cell lines. It works by blocking FGFR autophosphorylation, disrupting signal transduction and the subsequent activation of downstream cellular signaling pathways. What sets pemigatinib apart from earlier FGFR inhibitors is its heightened selectivity for members of the FGFR family.

Its effectiveness and pharmacological properties have allowed it to demonstrate antitumor activity at lower doses in preclinical settings. Pemigatinib is approved for use as a standalone treatment in adult patients with locally advanced or metastatic cholangiocarcinoma that exhibits FGFR2 fusion or rearrangement, particularly in cases where patients have relapsed or are unresponsive after at least one prior systemic therapy. However, its clinical efficacy can be compromised by the development of acquired resistance, primarily through secondary mutations in the kinase domain of FGFR2 [18]. Numerous clinical trials are currently underway exploring the use of pemigatinib in various cancers, including bladder cancer, nonmuscle invasive bladder cancer, recurrent urothelial carcinoma (NCT03914794), pancreatic cancer (NCT05216120), urothelial cancer (NCT04294277), gastrointestinal cancer (NCT05651672), non-small cell lung cancer (NCT05210946), breast cancer (NCT05560334), and, most recently, recurrent GBM and other primary central nervous system tumors with an activating FGFR1-3 mutation or fusion/rearrangement (NCT05267106).

Based on all the previous preclinical and clinical data, the main objective of our study is to evaluate the radiosensitizing potential of pemigatinib, which specifically targets FGFR1, for differentiated glioblastoma cells and GSCs, when used in conjunction with radiotherapy both in vitro and in vivo.

## 2. Materials and Methods

### 2.1. Cell Culture

This study utilized two types of GBM cells: differentiated cells and stem cells. Differentiated human GBM cells, U87MG (RRID:CVCL_0022) (MGMT methylated) and LN-18 (RRID:CVCL_0392) (MGMT unmethylated) from the American Type Culture Collection (Rockville, MD, USA), were cultured according to methods outlined in a previous publication [10].

Primary stem cell lines, named GC1 and GC2, were derived from GBM biopsy specimens obtained from the Neurosurgery Department of Toulouse University Hospital as part of the clinical study STEMRI, led by PI Pr E. Cohen-Jonathan Moyal. This study received approval from the Human Research Ethics Committee (Ethics Code 12TETE01, ID-RCB No. 2012-A00585-38, Approval Date: 7 May 2012), and the results have recently been published [19]. Written informed consent was secured from all the participating patients. The World Health Organization classified all the tumors involved as GBM. The characteristics of GC1 and GC2 as stem cells have previously been detailed by our group [13]. The cells were routinely tested for mycoplasma using MycoAlert (Lonza, Basel, Switzerland). GC1 and GC2 were previously characterized as glioblastoma stem cell lines [13] and cryopreserved at early passages (P2–P3). To preserve their stem-like characteristics, all the experiments in this study were conducted using cells between passages 3 and 10 after initial derivation from patient tumor tissue.

### 2.2. Reagents

Pemigatinib (Pemazyre^®^, INCB054828) was obtained from Incyte and dissolved in DMSO (in vitro) or in 5% DMAC, 50 mM citrate buffer, and 0.5% methyl cellulose (in vivo). MG132 was diluted in ethanol and used at 20 µM. All the reagents, except pemigatinib, were purchased from Sigma-Aldrich (Sigma, Saint-Quentin Fallavier, France).

### 2.3. Cell Viability

GC1 and GC2 cells (150,000/well, 24-well plates) were treated with pemigatinib for 24–48 h at 125, 250, and 500 nM. The cell number was determined post-treatment, after the dissociation of neurospheres, using a Bio-Rad counter (Biorad, Hercules, CA, USA). U87 and LN18 cells (10,000/well, 96-well plates) were treated with pemigatinib at 1, 2, and 3 μM, and viability was assessed via the MTT assay (Abcam, Cambridge, UK). For combination experiments with temozolomide (100 μM) in differentiated GBM cells, 1 μM pemigatinib was chosen as the lowest concentration that produced a detectable biological effect in these cells, enabling the assessment of potential interaction with TMZ while minimizing off-target effects.

### 2.4. siRNA Transfection

Neurospheres were dissociated and plated at 500,000 cells in 5 mL of culture medium 24 h before transfection. Transfection was carried out using Lipofectamine RNAimax (Invitrogen, Waltham, MA, USA) according to the manufacturer’s instructions. siRNAs targeting *FGFR1* (SI03094637), *S100A4*(1) (SI00709667), *S100A4*(8) (SI04227916), and a scramble control siRNA (siscr) were acquired from Qiagen (Qiagen, Venlo, The Netherlands).

### 2.5. Quantitative Real-Time PCR

For the quantitative analysis of gene expression for the human *FGFR1*, *FOXM1*, *S100A4*, and glyceraldehyde-3-phosphate dehydrogenase (*GAPDH*) genes, total RNA was extracted using the RNeasy RNA isolation kit (Qiagen), reverse-transcribed using the PrimeScript RT Reagent Kit (TAKARA, Kusatsu, Japan), and amplified using SsoFast EvaGreen Supermix (Bio-Rad, Marnes-la-Coquette, France). Gene expression was normalized to GAPDH using StepOne+ (Applied Biosystems, Waltham, MA, USA).

### 2.6. Western Blot

Proteins were extracted in 50 mM Tris-HCl (pH 7.5), 0.1% Triton X-100, 5 mM EDTA, and protease inhibitors. SDS-PAGE was performed using Mini-PROTEAN TGX Stain-Free Gels (Bio-Rad) followed by transfer to nitrocellulose membranes (Trans-Blot Turbo Transfer Packs, Bio-Rad) using the Trans-Blot Turbo Transfer system (Bio-Rad). The membranes were incubated with the following primary antibodies: anti-FGFR1 (D8E4) XP rabbit (1:1000 dilution; Cat# 9740, Cell Signaling Technology, Danvers, MA, USA), anti-FRS2 (EPR14724) (1:1000 dilution; abcam, Cambridge, UK), anti-actin clone C4 (1:20,000 dilution; Cat# MAB1501, Millipore, Molsheim, France), and anti-phospho-FRS2-alpha (Tyr196) (1:1000 dilution; Cell Signaling Technology Cat# 3864). Signals were visualized with ECL kits (ECL RevelBlot Plus and ECL RevelBlot Intense, Ozyme, Saint-Cyr-l’école, France) and imaged with Chemi-Doc (Bio-Rad).

### 2.7. Flow Cytometry

GSCs treated with 500 nM pemigatinib for 48 h were fixed and permeabilized (Cytofix/Cytoperm, BD Biosciences, Franklin Lakes, NJ, USA). The cells were then washed and incubated for 30 min at 4 °C in PBS supplemented with 10% FBS to block nonspecific binding. Subsequently, the cells were incubated for 1 h at 4 °C in the dark with an S100A4-PE conjugated primary antibody (NBP2-54580APC, NOVUS Biologicals, Cambridge, UK) or a mouse IgG1 kappa isotype control (17-4714-42, Invitrogen).

Data were acquired using a MACSQuant10 (Miltenyi Biotec, Bergisch Gladbach, Germany) and analyzed in FlowJo^TM^ v10.9 Software (BD Life Sciences, Franklin Lakes, NJ, USA). The S100A4 levels were quantified as the specific fluorescence index (SFI) = (Geomean antibody − Geomean IgG control)/Geomean IgG control [20].

### 2.8. Clonogenic Assay

U87 and LN18 cells (500/well, 6-well plates) were treated with irradiation, TMZ, and/or pemigatinib. After one week, colonies were stained with crystal violet. The survival fraction was calculated using the following formula: Survival fraction = (colonies counted/(cells seeded × plating efficiency)) × 100. Here, plating efficiency (PE) is the seeding efficiency, defined as the ratio of the number of colonies formed to the number of cells initially seeded.

Primary neurospheres treated with pemigatinib or siRNA targeting *S100A4* or vehicle were seeded in 96-well plates at a concentration of 500 cells/well, with 12 wells per condition. The following day, the cells were irradiated or not irradiated with X-rays ranging from 2 to 6 Gy (SmART+ irradiator, Precision X-ray Inc., Madison, CT, USA). One week post irradiation, the number of neurospheres composed of more than 20 cells was assessed. The surviving fraction was determined according to the PE concentration under nonirradiated conditions. PE is defined as follows: PE = number of spheres/number of seeded cells × 100. The sphere size was analyzed using Fiji (ImageJ version 1.53q, Java 1.8.0_172, 64-bit).

### 2.9. RNA Sequencing

For transcriptomic analysis, the GC1 and GC2 glioblastoma stem cell lines were seeded at 2 × 10^6^ cells in 5 mL of medium in 25 cm^3^ flasks and treated with either 250 nM pemigatinib or DMSO control for 48 h (three independent biological replicates per condition per cell line; *n* = 6 total per condition). RNA was extracted using the RNeasy kit (Qiagen) and quality verified (RQN > 8) prior to library preparation with the Illumina Stranded Total RNA Prep protocol, including rRNA depletion, strand-specific library construction, and paired-end sequencing. The library quality and concentration were assessed using the Qubit™ dsDNA BR Assay (Thermo Fisher Scientific, Waltham, MA, USA), High Sensitivity NGS Fragment Analysis Kit (Agilent Technologies, Santa Clara, CA, USA), and KAPA Library Quantification Kit for Illumina^®^ (Roche, Basel, Switzerland). Sequencing was performed on an Illumina NextSeq 550 (San Diego, CA, USA) with a High Output flow cell, and the raw reads were demultiplexed with bcl2fastq v2.20.0.422. The transcriptomic results represent an integrated differential expression analysis across both models, averaged to highlight changes conserved between GSC lines. The RNA-seq data have been deposited in the NCBI SRA under accession PRJNA1320628.

### 2.10. Fusion and Mutation Determination

*Reference and alignment*. Reads were aligned to GRCh38 using STAR v2.7.x in two-pass mode with default gene model guidance (GTF: Ensembl release v98). Soft clipping and chimeric junction output were enabled for fusion discovery.

*Fusion calling*. Gene fusions were called with Arriba v2.x (using the recommended blacklist and database bundles) [21] and STAR-Fusion v1.x (with the CTAT resource library built on GRCh38; Ensembl v98) [22]. The minimal evidence requirements followed the tool defaults (e.g., Arriba: ≥1 junction read; STAR-Fusion: junction + spanning support). The results were filtered to remove read-through events, paralog artifacts, mitochondrial fusions, and panel-of-normal hits. High-confidence fusions were defined as (i) detected by both callers or (ii) single-caller calls with strong supporting evidence and manual confirmation in IGV. Under these criteria, no high-confidence FGFR1/2/3 fusions were detected in GC1 or GC2.

*Variant calling from RNA-seq*. Putative variants in FGFR1–3 were identified from RNA-seq using GATK HaplotypeCaller (gatk-4.2.0.0, GRCh38v98, https://gatk.broadinstitute.org/hc/en-us/articles/360037225632-HaplotypeCaller (accessed on 1 September 2025)) following RNA-seq best practices: SplitNCigarReads, base quality recalibration, and variant calling restricted to the coding regions of canonical transcripts (Ensembl v98). Variants were filtered with hard thresholds (e.g., low depth/quality or strand bias) and annotated using VEP (SIFT/PolyPhen) with cross-reference to ClinVar, COSMIC, and CIViC. Variants were categorized as pathogenic/likely pathogenic, benign/likely benign, or VUS. Consistent with the results, only VUS (and known SNPs) were found in FGFR1–3; no activating kinase-domain mutations were identified. Selected loci were reviewed visually in IGV to confirm read support and rule out mapping artifacts. This analysis was conducted using Alamut Visual software (version 2.11-0; Interactive Biosoftware, Rouen, France).

### 2.11. Bioinformatic Analysis

*QC and pre-processing*. The FASTQ quality was assessed with FastQC and summarized with MultiQC (per-base quality, adaptor content, GC distribution). The post-alignment QC included the mapping rate, multimapping, rRNA content, gene body coverage, strandedness, and duplication estimates (Picard).

*Quantification and DE*. Gene-level counts were generated with featureCounts (union exon model; stranded = reverse for the Illumina stranded protocol). Differential expression was assessed in R/DESeq2 with the design~cell_line + treatment to estimate the treatment effect while accounting for GC1/GC2 differences. Multiple testing was controlled by the Benjamini–Hochberg procedure; unless noted, genes with adjusted *p* < 0.05 and |log_2_FC| > 1 were considered significant. The log_2_FC values represent line-averaged effects when regulation was concordant across GC1 and GC2.

*Pathway and enrichment analyses*. The mRNA expression levels of FGFR receptors and S100A4 in GBM tissue compared to normal tissue were evaluated using Gene Expression Profiling Interactive Analysis (GEPIA, http://gepia.cancer-pku.cn/ (accessed on 1 September 2025)), a web-based tool that provides fast and customizable analyses based on TCGA and GTEx data. For this analysis, the log_2_ fold change (log_2_FC) cutoff was set at 1, and the *p*-value cutoff was set at 0.01. The gene expression values are presented as log_2_(TPM + 1), where TPM denotes transcripts per million.

A volcano plot was generated to visualize the differential gene expression in GSCs treated with or without pemigatinib, using RNA-seq data and the Srplot tool (Scientific and Research plot tool, http://www.bioinformatics.com.cn/SRplot (accessed on 1 September 2025)). This free online platform was also used to visualize the principal pathways affected in GSCs treated with pemigatinib. Impacted and downregulated genes were identified through Gene Ontology (GO) analysis using the web-based portal Metascape (https://metascape.org (accessed on 1 September 2025)).

### 2.12. In Vivo Experiments

Six-to-eight-week-old female nude mice were used in accordance with a protocol (APAFIS# 24969-2020040210408334 v4; approval date: 11 June 2020) established by the Institutional Animal Care and Ethic Committee (UMS006 CEEA-122; President: Nicolas Cenac). The NMRI nu/nu mice (Janvier Labs, Genest Saint Isle, France) were orthotopically implanted with 250,000 GC1 cells that had been previously transduced with a vector containing the luciferase gene, GFP gene, and geneticin resistance gene (#LVP403, Amsbio, Abingdon, UK), under approval #2621 for the contained use of genetically modified organisms for research purposes. Tumor progression was monitored weekly using IVIS bioluminescence imaging. Before the start of treatment (day 23), tumor monitoring using IVIS imaging was performed to identify the mice in which tumor cell implantation successfully led to tumor formation. Each group initially included 20 mice. Mice that did not develop tumors were excluded from the study based on the IVIS imaging results. Specifically, 7 mice were excluded from both the control group and the pemigatinib-treated group, 3 mice from the irradiated group, and 6 mice from the group receiving both pemigatinib and irradiation. To ensure comparability between experimental groups, only mice with equivalent tumor sizes were included in each group. Beginning 23 days after xenografting, the mice were administered pemigatinib (0.5 mg/kg) by gavage 5 days a week for the duration of the experiment, until 36 days post-xenografting. The mice received localized irradiation at a dose of 5 Gy (SmART+ irradiator, Precision X-ray Inc., Madison, CT, USA). They were sacrificed as soon as clinical symptoms appeared. To ensure consistency, all animals were treated and measured in the same order by the same experimenters. In addition, different experimenters were responsible for gavage administration and tumor monitoring to reduce the potential bias associated with individual handling or measurement.

### 2.13. Statistical Analysis

Data are presented as mean ± SD. Comparisons used Student’s *t*-test or one-way ANOVA. Kaplan–Meier survival was analyzed using the log-rank test (GraphPad Prism version 10.1.2, GraphPad Software, Boston, MA, USA).

## 3. Results

### 3.1. Expression of FGF Receptors in GBM and GBM Cells

GEPIA showed that FGFR1 was the only receptor significantly overexpressed in GBM compared to normal brain tissue (Figure 1). As our previous research demonstrated, FGFR1 was the most highly expressed FGFR receptor in differentiated U87 [23] and LN18 [10] cells, as in patient-derived GSCs (GC1, GC2) [13]. Based on these findings, we focused our investigation on the effects of pemigatinib on FGFR1.

### 3.2. Radiosensitizing Effect of Pemigatinib on Differentiated MGMT-Methylated and MGMT-Unmethylated GBM Cells

We investigated the radiosensitizing effect of pemigatinib on two radioresistant GBM cell lines with different MGMT methylation statuses: U87 (MGMT-methylated) and LN18 (MGMT-unmethylated). We first evaluated the impact of pemigatinib on FGFR1 signaling by assessing the phosphorylation of FRS2, a direct downstream target. U87 and LN18 cells were treated with increasing concentrations of pemigatinib for 1 and 24 h. In both lines, a dose-dependent reduction in P-FRS2 was observed at concentrations ranging from 1 to 3 µM (Figure 2A). Despite the effective inhibition of FRS2 phosphorylation, cell viability was not significantly affected after 24 or 48 h of treatment (Figure 2B). We then assessed the functional effects of pemigatinib using clonogenic assays, alone and in combination with standard therapies. In MGMT-methylated U87 cells, pemigatinib significantly reduced colony formation with or without radiation. As expected, TMZ alone almost completely suppressed colony formation in this TMZ-sensitive line. However, no additional effect was seen when combining pemigatinib with TMZ or radiation plus TMZ (Figure 2C). In contrast, in MGMT-unmethylated LN18 cells, TMZ had no impact on colony formation. Pemigatinib alone reduced colony formation by 50%, and the combination of pemigatinib and TMZ reduced it further (66%). Radiation slightly enhanced TMZ effects, but only the combination of pemigatinib and TMZ showed clear radiosensitization, indicating potential in TMZ-resistant models.

### 3.3. Effect of Pemigatinib on FGFR1 in GSCs

To assess the impact of pemigatinib on GSCs, we utilized two primary GSC lines, GC1 and GC2, derived from surgical GBM samples from different patients. Unlike in differentiated cells, the inhibition of phosphorylated FRS2 expression in GSCs was observed at lower concentrations of pemigatinib, ranging from 125 to 500 nM, after 24 and 48 h of treatment. Notably, while pemigatinib is known as a tyrosine kinase inhibitor, it also caused a significant reduction in FGFR1 expression in both GC1 and GC2 cells at these time points (Figure 3A). We hypothesized that the observed reduction in FGFR1 expression could be due to FGFR1 internalization followed by proteasome-dependent degradation. To investigate this, cells were treated with MG-132, a potent inhibitor of proteasome-mediated intracellular protein degradation. As shown in Figure 3B, MG-132 treatment prevented the pemigatinib-induced decrease in FGFR1 expression, confirming its degradation via the proteasome.

Next, we evaluated the effect of pemigatinib on the viability of GC1 and GC2 cells. As illustrated in Figure 3C, treatment with increasing concentrations of pemigatinib (125, 250, and 500 nM) led to a dose-dependent reduction in cell survival in both GSC lines at 24 and 48 h. In GC1 cells, all the tested concentrations significantly decreased cell survival compared to control, with the most pronounced effect observed at 500 nM (*** *p* < 0.001 at 24 h; ** *p* < 0.01 and *** *p* < 0.001 at 48 h). Similarly, in GC2 cells, pemigatinib induced a significant reduction in cell viability at all concentrations and time points, with effects becoming more robust at higher doses and extended exposure times (** *p* < 0.01 to **** *p* < 0.0001). These findings indicate that pemigatinib reduces GSC survival in a concentration- and time-dependent manner, suggesting its potential therapeutic efficacy against FGFR-driven tumor growth.

### 3.4. FGFR1 Inhibition by Pemigatinib Reduces Neurosphere Formation and Enhances GSC Radiosensitization

Neurosphere assays showed that pemigatinib significantly reduced the sphere-forming capacity in both GC1 and GC2 cells in a dose-dependent manner. In GC1 cells, treatment with 250 nM and 500 nM pemigatinib led to a marked decrease in the number of spheres formed compared to the control group (** *p* < 0.01, *** *p* < 0.001). Similarly, GC2 cells exhibited a significant and dose-dependent reduction in sphere formation following treatment with pemigatinib, with both concentrations showing highly significant effects (**** *p* < 0.0001). These findings suggest that pemigatinib effectively impairs the self-renewal capacity of GSCs, highlighting its potential to target the tumor-initiating cell population in GBM. Clonogenic survival assays were then performed to assess the impact of pemigatinib on the radiosensitivity of the GSC lines GC1 and GC2. In both cell lines, irradiation alone reduced survival compared to nonirradiated controls. However, the limited radiosensitizing effect of pemigatinib observed in GC1 cells (Figure 3E) may be explained by its strong effect as a monotherapy, as shown in the sphere formation assay (Figure 3D). The substantial reduction in the baseline survival of GSCs was already markedly decreased prior to irradiation, potentially masking any additive or synergistic effect of combined treatment with radiation. In contrast, GC2 cells exhibited a notable enhancement in radiosensitivity following treatment with 500 nM pemigatinib, particularly at higher radiation doses (6–8 Gy), where a statistically significant reduction in the survival fraction was observed compared to that with radiation alone (** *p* < 0.01) (Figure 3E) despite the fact that pemigatinib also reduced sphere formation in GC2 cells (Figure 3D). These findings suggest that the radiosensitizing potential of pemigatinib may depend on the intrinsic sensitivity of individual GSC populations to FGFR inhibition, with variations in response likely reflecting underlying biological differences between GSC subtypes. As shown in Figure 3F, pemigatinib significantly reduced the size of GSC spheres in both GC1 and GC2 lines, under both basal (0 Gy) and irradiated (4 Gy) conditions. In GC1 cells, sphere size was markedly reduced by pemigatinib alone at both 250 nM and 500 nM (**** *p* < 0.0001), and this reduction persisted following irradiation, with no clear additive effect. This is consistent with prior observations in Figure 3D, where pemigatinib monotherapy strongly impaired sphere-forming ability, and with Figure 3E, which shows limited radiosensitization in GC1, likely due to an already strong effect of pemigatinib as a single agent. In contrast, GC2 cells displayed a more moderate reduction in sphere size in response to pemigatinib alone (** *p* < 0.01), but a further decrease was observed when it was combined with 4 Gy radiation (** *p* < 0.01 to *** *p* < 0.001). This is consistent with Figure 3D,E, which illustrates that GC2 cells showed partial sensitivity to pemigatinib alone, and increased radiosensitivity when pemigatinib was combined with radiation, especially at higher doses. In this context, the sphere size at the end of the clonogenic assay was used as an indicator of the proliferative capacity of cells within each sphere, complementing the survival fraction data. Together, these results reinforce the idea that the radiosensitizing potential of pemigatinib is context-dependent and more apparent in GSC populations like GC2, where its monotherapy effect is moderate and leaves room for additive interactions with radiation.

### 3.5. Pemigatinib Efficacy in TMZ-Resistant GSCs

Pyrosequencing confirmed that both GC1 and GC2 were MGMT-unmethylated, explaining their resistance to TMZ [20]. As expected, TMZ did not reduce sphere formation or enhance radiosensitivity in GC2 (Appendix A). In GC2 cells, pemigatinib alone or in combination with TMZ appeared to reduce sphere formation, with this trend being maintained under increasing doses of irradiation. These supplemental data illustrate the potential effect of pemigatinib in TMZ-refractory GSCs and warrant further validation.

### 3.6. FGFR Genetic Profiling

Since the literature has described pemigatinib as effective for GBM patients with fusions/rearrangements or mutations in FGFRs, we investigated whether the GC1 and GC2 cells used in this study have these characteristics. With respect to the RNA sequencing data obtained from GC1 and GC2, gene fusions were detected via the bioinformatics pipelines Arriba and STAR. Our analysis revealed no fusion of any FGF receptor in GC1 or GC2 cells.

In the GC1 cell line, alterations in the *FGFR1*, *FGFR2*, and *FGFR3* genes were identified. Specifically, for *FGFR1*, three mutations were classified as variants of unknown significance (VUS), and one was a single-nucleotide polymorphism (SNP) with no structural impact on the protein. *FGFR2* exhibited only one mutation, a VUS in exon 13. For *FGFR3*, two VUS mutations and two mutations not listed in the consulted databases were found. In the GC2 line, alterations in the *FGFR3* gene were primarily detected as VUS in exon 15 of the tyrosine kinase domain, with one mutation in *FGFR1* and no mutations in *FGFR2* (Appendix A). VUS are often so rare that limited information is available, making it difficult to determine whether the variant is disease-related or not.

### 3.7. The Effect of Pemigatinib on the Gene Expression of GSCs

To explore the molecular mechanisms underlying the effects of pemigatinib on GSCs, transcriptomic profiling was performed to compare gene expression between pemigatinib-treated and control cells. The results presented are compiled from data obtained from GC1 and GC2 cells treated with pemigatinib. As shown in the volcano plot (Figure 4A), pemigatinib induced significant differential expression of a large number of genes, with 862 genes downregulated and 612 upregulated (adjusted *p* < 0.05, |log_2_FC| > 1). Among the most significantly downregulated genes were *FGFR1*, the primary target of pemigatinib, along with the oncogenic regulators *S100A4* and *FOXM1*, both of which are associated with GSC maintenance, proliferation, and resistance to therapy—findings that are consistent with our previous results obtained using *FGFR1*-targeting siRNA in GC1 and GC2 cells [2,13].

Gene Ontology analysis revealed the downregulation of pathways related to mitotic cell cycle, chromatin organization, DNA repair, and radiation response (Figure 4B). These findings suggest that FGFR inhibition disrupts key molecular programs required for proliferation and resistance to DNA damage, consistent with the observed effects on sphere formation, size reduction, and radiosensitization in GSC models (see Figure 3D,F). Notably, the downregulation of DNA repair and radiation response pathways provides mechanistic support for the enhanced radiosensitivity observed in GC2 cells. To assess the clinical relevance of *S100A4*, one of the most strongly downregulated genes following pemigatinib treatment, we analyzed its expression in GBM patient samples. The expression of *S100A4* was significantly elevated in GBM tissues compared to normal brain (*p* < 0.05), consistent with its established roles in tumor progression, stemness, and resistance to therapy (Figure 4C). The downregulation of *S100A4* by pemigatinib suggests a potential mechanism by which FGFR inhibition may suppress aggressive and treatment-refractory phenotypes in GBM. Notably, S100A4 has been identified as both a biomarker and functional regulator of GSCs, and as a key mediator of mesenchymal transition and stemness in GBM [24].

### 3.8. S100A4 Downregulation Contributes to Enhanced Radiosensitivity in GSCs

To further investigate the functional relevance of *S100A4* downregulation following *FGFR* inhibition, gene expression was first validated in GC1 and GC2 cells treated with pemigatinib. As shown in Figure 5A, pemigatinib significantly reduced the expression of *FGFR1*, *FOXM1*, and *S100A4* in both GC1 and GC2 cells (*** *p* < 0.001 to **** *p* < 0.0001), consistent with transcriptomic data (Figure 4A). The quantification of S100A4 protein levels by FACS analysis confirmed that pemigatinib significantly decreased S100A4 expression in both cell lines compared to control (* *p* < 0.05; Figure 5B). To directly assess the contribution of S100A4 to cell survival and the radiation response, siRNA-mediated knockdown of *S100A4* was performed in GC1 and GC2 cells. Efficient suppression of *S100A4* mRNA was confirmed using two independent siRNAs in both cell lines (**** *p* < 0.0001; Figure 5C). Flow cytometry analysis confirmed efficient S100A4 silencing at the protein level, showing a 37–49% reduction in GC1 and a 55% reduction in GC2 compared with siCtrl, in line with the mRNA expression data (Figure 5D). Functional assessment using clonogenic survival assays revealed that S100A4 silencing significantly sensitized both GC1 and GC2 cells to ionizing radiation across multiple doses (2–6 Gy) compared to scrambled siRNA controls (* *p* < 0.05 to ** *p* < 0.01; Figure 5E). These findings demonstrate that S100A4 plays a key role in mediating radioresistance in GSCs and support the hypothesis that its downregulation by FGFR inhibition contributes to the radiosensitizing effects of pemigatinib observed in GC2 cells.

### 3.9. In Vivo Study of the Therapeutic Efficacy of Pemigatinib for Human GSC Orthotopic Xenografts

To evaluate pemigatinib’s therapeutic efficacy in vivo, we used an orthotopic GBM model with GC1 cells transduced with a luciferase reporter, allowing weekly tumor monitoring via IVIS bioluminescence imaging (Figure 6A). This ensured a consistent tumor burden at the treatment start. Mice received pemigatinib (0.5 mg/kg, oral gavage) from day 23 post-implantation, five times per week. A single 5 Gy dose of cranial radiation was administered on day 36. The animals were monitored daily and euthanized upon the appearance of neurological symptoms or other clinical endpoints. Body weight was monitored weekly for the first 3–4 weeks. The trajectories (% change from baseline; mean ± SD) were comparable between groups, without an early weight-loss signal in the pemigatinib or combination arms (Appendix A).

Survival analysis revealed that both monotherapies significantly improved overall survival compared to untreated controls. The median survival was 77 days in the control group, 118.5 days in the irradiation-only group (*p* = 0.0106 vs. control), and 140.7 days in the pemigatinib-only group (*p* = 0.0029 vs. control) (Figure 6B,C). The combination of pemigatinib and irradiation resulted in the greatest survival benefit, with a median survival of 177.7 days (**** *p* < 0.0001 vs. control). However, no statistically significant differences were observed when comparing the combination group to either monotherapy (*p* > 0.4), suggesting that the therapeutic effects of pemigatinib and irradiation may not be synergistic in this GC1-driven model.

## 4. Discussion

Despite the significant improvement in patient survival achieved with the introduction of TTFields therapy combined with temozolomide and radiotherapy as a standard treatment [25], GBM remains aggressive and prone to recurrence, emphasizing the need for new therapeutic strategies. In this study, we evaluated pemigatinib, a selective FGFR inhibitor, for its radiosensitizing and antiproliferative effects in GBM. Based on GEPIA and prior data, FGFR1 emerged as the most overexpressed FGFR in GBM and was consistently detected in all the models tested. This led us to focus on the consequences of FGFR1 inhibition. Our previous research demonstrated that FGFR1 inhibition in both differentiated cells and GSCs induced a radiosensitizing effect [10,13]. Additionally, we demonstrated that pemigatinib effectively inhibited FGFR1 signaling by reducing FRS2 phosphorylation and promoting FGFR1 degradation via the proteasome pathway. While short-term viability assays in differentiated GBM cells showed minimal effects, likely due to cell adhesion-mediated resistance via focal adhesion kinase [26,27], clonogenic assays revealed a pronounced reduction in colony formation particularly in the MGMT-unmethylated LN18 line. In this TMZ-resistant context, pemigatinib alone reduced colony formation by 50%, and its combination with TMZ and radiation further enhanced radiosensitivity. Although we did not formally assess drug synergy, using 1 μM pemigatinib, the lowest active concentration in differentiated cells, enabled us to test whether modest FGFR1 inhibition could influence the TMZ response, particularly in MGMT-unmethylated models, suggesting therapeutic potential in patients with limited response to alkylating agents. Importantly, this therapeutic effect occurred despite the absence of FGFR gene fusions or known activating mutations in the LN18, suggesting that functional FGFR1 pathway activity, rather than mutational status alone, may be sufficient to confer sensitivity to FGFR inhibition. This highlights a potential shift in biomarker strategies, moving from genomic profiling toward assessing pathway dependency, broadening the scope of patients who may benefit from FGFR-targeted therapy. In GSCs, pemigatinib exhibited potent antitumor activity, significantly reducing neurosphere number and size in a dose-dependent manner and inducing significant cell death at nanomolar concentrations. While differentiated GC1 and GC2 cells could theoretically serve as more direct comparators, our preliminary experiments showed that FGFR1 expression was not significantly altered by pemigatinib at concentrations effective in the stem cell state (125–500 nM). As shown in Appendix A, FGFR1 protein levels remained unchanged in differentiated GC1 and GC2 cells, suggesting that FGFR1 pathway activity is reduced or less accessible in the differentiated state. Radiosensitization by pemigatinib was found to be context-dependent: GC1 cells, which were highly sensitive to pemigatinib monotherapy, showed limited additive benefit when pemigatinib was combined with irradiation. In contrast, GC2 cells, which had moderate single-agent sensitivity, displayed synergistic radiosensitization, particularly at higher radiation doses, as reflected in the reduced sphere size and survival fraction. The pronounced antiproliferative effect of pemigatinib appears to be mediated through the disruption of cell cycle regulation, as supported by our transcriptomic analysis. Other studies have also suggested that pemigatinib-induced G1 arrest is a common cytostatic mechanism across various FGFR expression patterns and tumor types [28], targeting pathways such as PI3K/AKT and RAF/MEK/ERK which are involved in cell growth, migration, proliferation, and metabolism [29,30].

Importantly, both GC1 and GC2 were found to be MGMT-unmethylated, a feature associated with poor response to standard therapy. The responsiveness of these cells to pemigatinib reinforces its potential as an alternative or complementary strategy in TMZ-refractory GBM.

Transcriptomic profiling further revealed that pemigatinib treatment downregulated genes involved in the mitotic cell cycle, DNA repair, and radiation response, providing mechanistic support for its radiosensitizing effects. Among the most significantly downregulated genes were *FGFR1*, *FOXM1*, and *S100A4,* each associated with GSC proliferation, mesenchymal transition, and resistance to therapy. S100A4, in particular, is a small calcium-binding protein that can act both extracellularly and intracellularly, affecting a variety of multiple biological processes depending on its binding partners. S100A4 is predominantly located in the cytoplasm and, to a lesser extent, in the nucleus. It interacts directly with proteins such as p53, annexin 2, and myosin IIA heavy chain, thereby enhancing apoptosis, cell migration, and angiogenesis [31,32]. S100A4 has been identified as an upregulator of ZEB1 and SNAIL2 and is implicated in mesenchymal transition in GSCs [24]. Furthermore, we demonstrated that FGFR1 and ZEB1 influence GBM cell proliferation and stemness [16]. These findings suggest a link between FGFR1, S100A4, and ZEB1, all associated with GSC proliferation, mesenchymal transition, and resistance to therapy. Our functional studies confirmed that S100A4 knockdown sensitized both GC1 and GC2 cells to irradiation, reinforcing its role as a key mediator of radioresistance. The inhibition of S100A4 by pemigatinib suggests that FGFR1 signaling modulates stemness and mesenchymal traits in GBM through downstream effectors such as S100A4 and ZEB1.

Together, these results suggest that pemigatinib can exert radiosensitizing effects at least in part through the suppression of stemness and mesenchymal programs driven by FGFR1 and its downstream effectors FOXM1 and S100A4. These mechanistic insights indicate that FGFR inhibition may complement existing therapeutic strategies in GBM, although further studies will be needed to confirm this in additional models. Importantly, this effect was observed despite the absence of FGFR gene fusions or known activating mutations in our models, suggesting that functional FGFR1 pathway activity, rather than mutational status alone, may contribute to sensitivity to FGFR inhibition. This raises the possibility of a shift in biomarker strategies, moving from purely genomic profiling toward assessing pathway dependency, thereby potentially broadening the scope of patients who could benefit from FGFR-targeted therapy. In addition to biomarker-driven selection, receptor crosstalk may also shape the response to FGFR inhibition. Although pemigatinib inhibits FGFR1–3, we centered our analyses on FGFR1 because protein-level profiling in patient-derived GC1/GC2 [33] showed robust FGFR1, no detectable FGFR2, and comparatively low FGFR3/FGFR4. Given the extensive RTK crosstalk in GBM, EGFR amplification/variants (e.g., EGFRvIII) may provide bypass signaling that limits the impact of FGFR1 blockade. Consistent with this, the feedback activation of EGFR upon FGFR inhibition and improved activity with dual FGFR–EGFR blockade have been reported [34], and receptor co-regulation at clathrin-coated sites supports direct EGFR–FGFR interplay at the plasma membrane [35].

An additional consideration is that FGFR3 expression was found to be higher in normal brain tissue compared to GBM (Figure 1). Since pemigatinib also targets FGFR3, potential effects on healthy tissue must be taken into account. Two factors may mitigate this concern. First, although pemigatinib is a potent inhibitor of FGFR1–3, it displays markedly reduced activity against the majority of non-FGFR kinases (>100-fold selectivity), thereby limiting broad off-target kinase inhibition. Second, while the blood–brain barrier (BBB) is frequently disrupted within GBM lesions, allowing drug penetration into tumor areas, it remains largely intact in surrounding non-tumoral regions. This differential permeability is likely to restrict drug access to FGFR3-expressing normal parenchyma. Consistent with this, clinical data from trials such as FIGHT-202 have reported primarily on-target adverse effects, including hyperphosphatemia due to altered phosphate handling, along with mucosal toxicities, nail and skin changes, and ocular events. These effects are manageable and largely reversible with supportive measures such as phosphate binders, local care, and dose adjustments [36,37]. Neurological adverse events have not been a prominent feature, suggesting limited CNS toxicity.

Another important issue is the possibility of acquired resistance to pemigatinib. In other cancers such as cholangiocarcinoma, resistance has been attributed to secondary mutations in the FGFR kinase domain or activation of alternative signaling pathways. Although such mechanisms have not yet been described in GBM, the intrinsic plasticity of glioblastoma stem cells and the frequent activation of bypass pathways (e.g., EGFR, MET, PDGFR) suggest that resistance could also emerge in this context. In light of our findings showing context-dependent radiosensitization, further studies will be needed to determine whether combining pemigatinib with radiotherapy or temozolomide could help prevent or delay resistance in a subset of GBM. Clinical evaluation will ultimately be required to clarify the efficacy and feasibility of such strategies.

Although a recent review [38] concluded that pemigatinib did not demonstrate efficacy in glioblastoma based on available clinical trials, it is important to note that this study primarily evaluated monotherapy, often in patients with previously treated recurrent GBM or other primary CNS tumors harboring FGFR1–3 mutations or fusions/rearrangements. Moreover, the molecular characterization of FGFR1–3 alterations has not been systematically compared between primary and recurrent tumors, which may differ significantly due to treatment-induced genetic and phenotypic changes. The ongoing phase II trial assessing pemigatinib (NCT05267106) also underscores the evolving landscape. Given the intrinsic complexity of GBM biology, the more aggressive and therapy-resistant nature of recurrent tumors, and the limited efficacy of monotherapies in this context, current conclusions may be premature and not definitive. Our findings suggest that FGFR1 inhibition, particularly in combination with radiotherapy, holds promise as a strategy for overcoming resistance mechanisms and improving therapeutic outcomes.

These findings are consistent with studies in other cancers, including triple-negative breast cancer, where the FGF2/FGFR1 transduction pathway induces the upregulation and secretion of S100A4 [39], with extracellular S100A4 released by tumor or stromal cells acting in an autocrine or paracrine manner by binding to its RAGE receptor, thus promoting cell migration, invasion, and angiogenesis [40,41]. Additionally, exosomal S100A4 plays a key role in hepatocellular carcinoma metastasis, by activating STAT3 phosphorylation and upregulating osteopontin expression [42], as well as inducing immunosuppression and non-small cell lung cancer development through STAT3 activation [43].

One limitation is the use of a limited number of primary GSC models. While GC1 and GC2 reflect clinically relevant MGMT-unmethylated GBM subtypes, additional studies across a broader panel of primary GSCs and patient-derived xenografts are needed to confirm the generalizability of these findings and to better define the subset of GBM patients most likely to benefit from FGFR1-targeted radiosensitization. Additionally, we did not include non-malignant CNS cells (e.g., human astrocytes or neural progenitor/neuronal models), which would allow the estimation of a therapeutic index. Clinical experience with pemigatinib indicates predominantly on-target, manageable toxicities without prominent neurological adverse events [36,37], but head-to-head assays in normal CNS models will be important to define selectivity. Although our study employed patient-derived glioblastoma stem cells and relied on functional assays to assess treatment impact, future studies incorporating flow cytometry-based analysis of stemness markers (e.g., Olig2, Sox2, Nestin) will be important to further characterize phenotypic shifts induced by FGFR inhibition and radiation.

Importantly, both GC1 and GC2 GSCs were MGMT-unmethylated and resistant to TMZ, yet still responded to pemigatinib. These findings suggest that FGFR inhibition may provide therapeutic benefit in a patient subset for whom standard-of-care alkylating agents are ineffective. Furthermore, neither GC1 nor GC2 harbored FGFR gene fusions or activating mutations typically associated with FGFR-targeted therapy in other cancers, though multiple variants of unknown significance (VUS) were present. This suggests that high FGFR1 expression and functional pathway activity may serve as a broader biomarker of FGFR dependency, independent of genomic alterations.

Finally, our in vivo data using an orthotopic GC1 xenograft model showed that both pemigatinib and radiation monotherapy significantly extended survival compared to controls. Although the combination did not statistically outperform either monotherapy, it achieved the longest median overall survival (177.7 days). This outcome, while not formally synergistic, is clinically meaningful, especially given the limited options for MGMT-unmethylated GBM. However, given the lack of statistical significance, we recognize that further studies are required to confirm the clinical relevance of this combined approach. The in vivo experiments were designed to assess therapeutic efficacy, and the observed survival benefit is interpreted in this context only, without inference for CSC dynamics.

Taken together, our results suggest that the combination of pemigatinib with radiotherapy, potentially alongside TMZ, may offer therapeutic benefit for patients with newly diagnosed GBM, particularly those with MGMT-unmethylated tumors and no detectable FGFR fusions or mutations. However, further preclinical and clinical studies are needed to confirm the efficacy of this approach.

## 5. Conclusions

This study demonstrates that FGFR1 inhibition with pemigatinib offers a promising strategy for overcoming the intrinsic radioresistance of GBM, including its most treatment-refractory subpopulations such as GSCs and MGMT-unmethylated tumors. FGFR1 inhibition not only enhances radiosensitivity but also downregulates key effectors such as S100A4, which are associated with tumor stemness, mesenchymal transition, and microenvironmental modulation. The survival benefit observed in orthotopic preclinical models underscores the translational relevance of combining pemigatinib with radiotherapy. Notably, this therapeutic efficacy occurred in the absence of FGFR fusions or activating mutations, suggesting broader applicability beyond genomically defined FGFR-driven tumors. Taken together, these findings support further preclinical and clinical evaluation of FGFR-targeted strategies as part of multimodal therapy in GBM, especially in treatment-refractory contexts.

## Figures and Tables

**Figure 1 cells-14-01427-f001:**
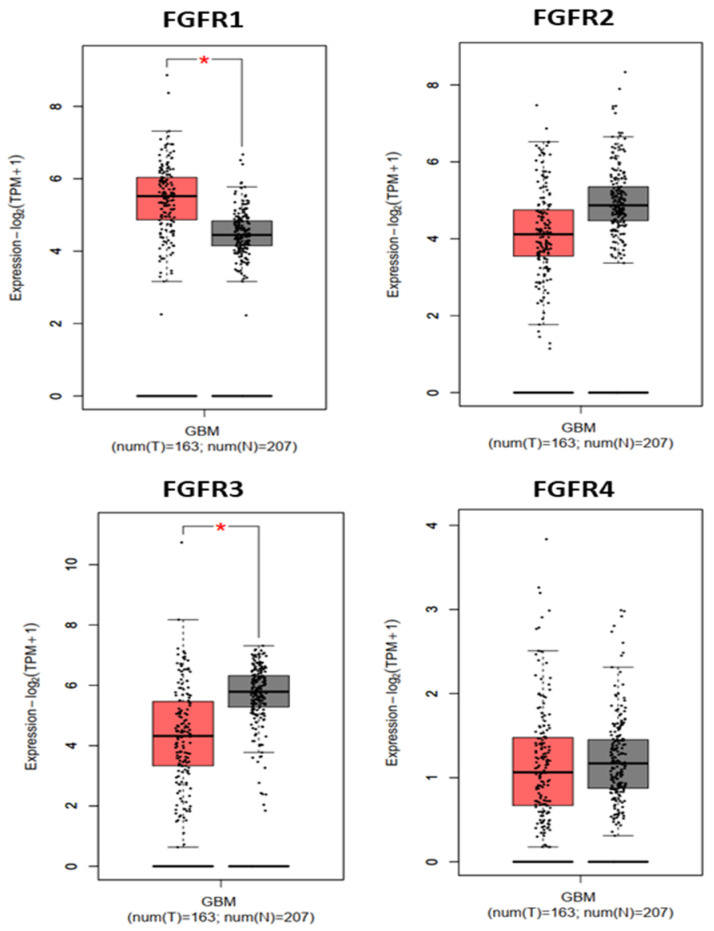
Expression levels of FGFR1–FGFR4 in glioblastoma (GBM) tumor tissue (T, *n* = 163) and normal brain tissue (N, *n* = 207), based on TCGA/GTEx RNA-seq datasets accessed via the GEPIA web server. Data are shown as log*_2_*(TPM + 1) values. Statistical comparisons were performed using one-way ANOVA; 0.01 < *p* < 0.05 is indicated by an asterisk.

**Figure 2 cells-14-01427-f002:**
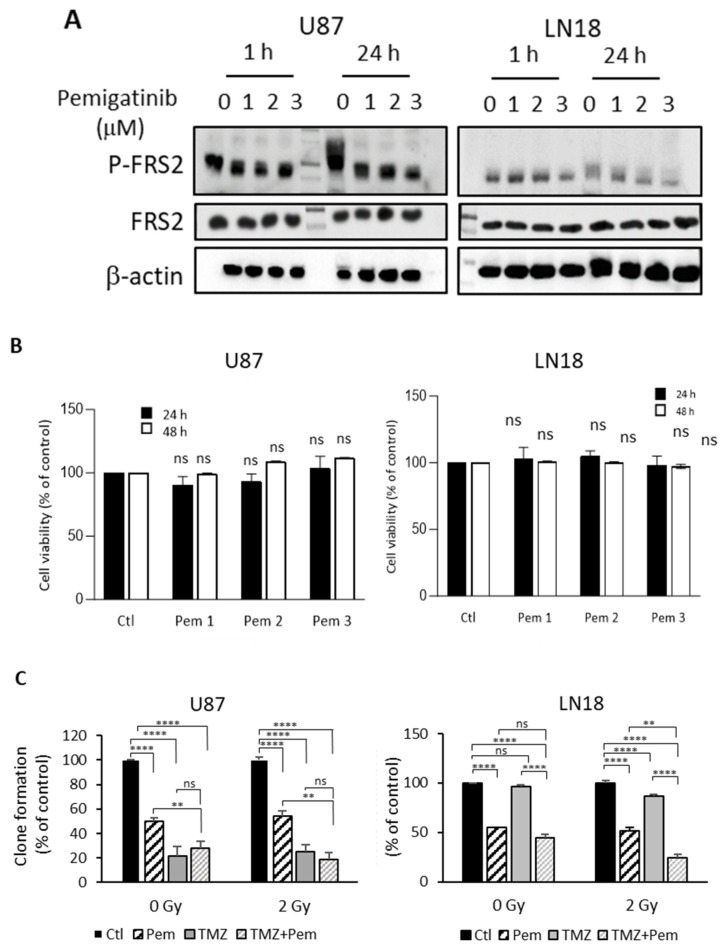
The impact of pemigatinib on differentiated glioblastoma cells. U87 and LN18 cells were treated with pemigatinib at concentrations of 1, 2, or 3 μM for 1 or 24 h. (**A**) Phospho-FRS2 and FRS2 levels were evaluated via Western blotting (representative of three independent experiments), with β-actin used for normalization. (**B**) Cell viability was measured using the MTT assay. (**C**) Clonogenic assays were performed on U87 and LN18 cells treated with pemigatinib (1 μM) and temozolomide (100 μM), with or without irradiation (2 Gy). The data for cell viability and clonogenic assays represent the quantification of three independent experiments, presented as the mean ± SD. ns *p* > 0.05, ** *p* < 0.01, **** *p* < 0.0001.

**Figure 3 cells-14-01427-f003:**
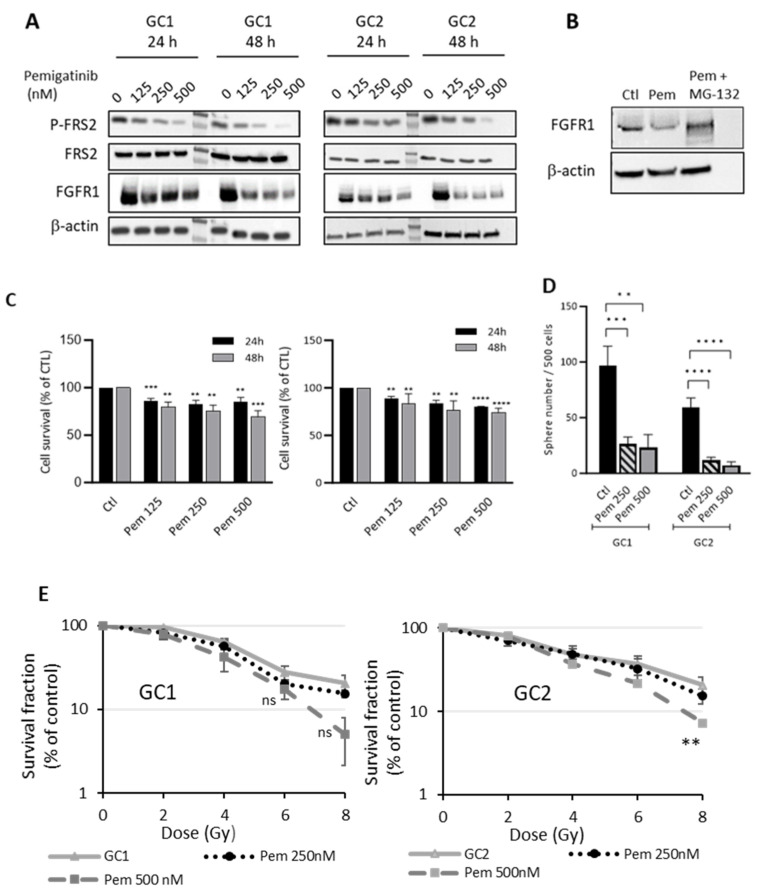
Effect of pemigatinib on glioblastoma stem cells (GSCs). Neurospheres derived from two patients (GC1 and GC2) were analyzed. (**A**) GC1 and GC2 cells were treated with pemigatinib for 24 h and 48 h, and phospho-FRS2 and FGFR1 expression was assessed via Western blotting; the Western blots are representative of three independent experiments. (**B**) GC1 cells treated with pemigatinib (250 nM) and MG132 (20 μM) for 48 h were analyzed for FGFR1 expression by Western blotting, and the Western blots are representative of three independent experiments. β-actin was used as a loading control. (**C**) GC1 and GC2 cells were treated with increasing concentrations of pemigatinib (125, 250, and 500 nM) for 24 h (black bars) or 48 h (gray bars). Cell survival was assessed by cell counting and is expressed as a percentage relative to untreated control cells (Ctl). (**D**) Sphere formation was evaluated in GC1 and GC2 cells treated with pemigatinib without irradiation, as described in the Methods section. (**E**) The survival fraction of spheres formed post-treatment with pemigatinib at 250 nM (dotted line) or 500 nM (dashed line) and increasing doses of ionizing radiation (2–8 Gy). The survival fraction was measured using a clonogenic assay and is expressed as a percentage of nonirradiated control cells. (**F**) GC1 and GC2 cells were subjected to a clonogenic assay and treated with pemigatinib (250 nM or 500 nM), with or without 4 Gy irradiation. At the end of the assay (day 10), the sphere size (in μm) was measured using Fiji software as an indicator of the proliferative capacity of the cells within each sphere. The data from panels (**C**–**F**) represent the mean ± SD of at least three independent experiments. Statistical significance was determined relative to control using *t* tests: * *p* ≤ 0.05, ** *p* < 0.01, *** *p* < 0.001, **** *p* < 0.0001.

**Figure 4 cells-14-01427-f004:**
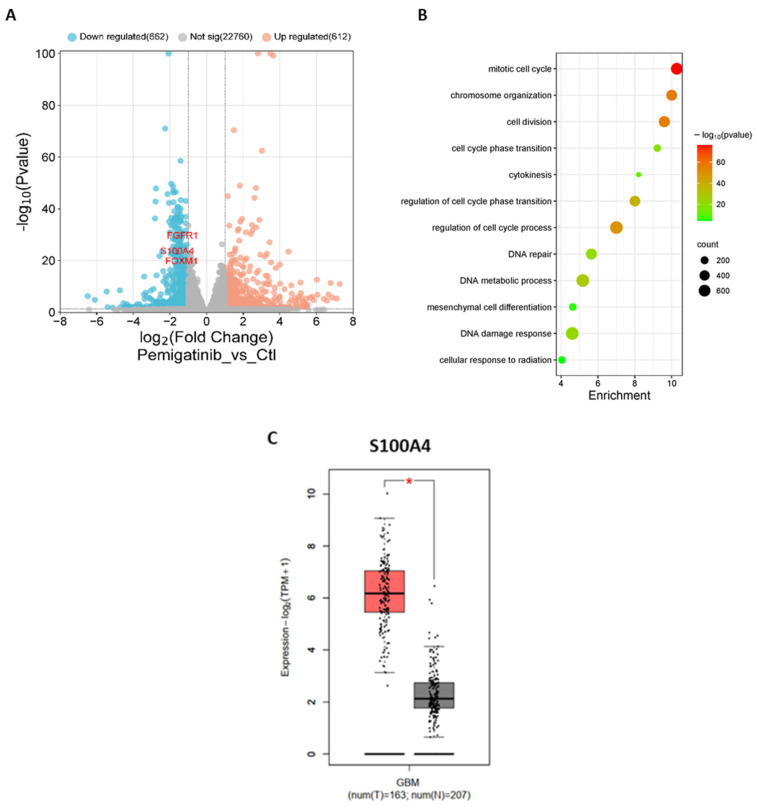
Analysis of genes and pathways affected by pemigatinib treatment. (**A**) Differential gene expression between glioblastoma stem cells (GC1 and GC2) treated with pemigatinib (250 nM, 48 h) and corresponding vehicle controls, based on an integrated analysis of both models (*n* = 3 biological replicates per condition per cell line). Data are presented as a volcano plot. (**B**) Gene Ontology analysis identified pathways downregulated by pemigatinib treatment. (**C**) Box plot comparing S100A4 expression in glioblastoma (GBM) tumor tissue (T, *n* = 163) versus normal brain tissue (N, *n* = 207) using TCGA/GTEx RNA-seq datasets via the GEPIA web server. Data are shown as log_2_(TPM + 1) values. Statistical analysis was performed using one-way ANOVA; *p* < 0.05 is indicated by an asterisk.

**Figure 5 cells-14-01427-f005:**
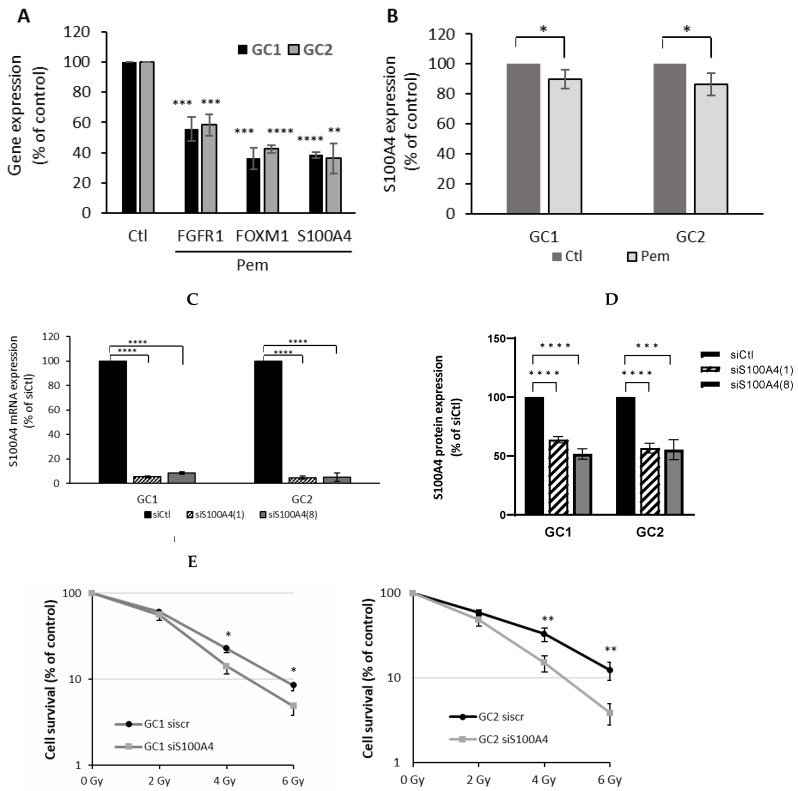
Effect of *S100A4* inhibition on the radiosensitization of GSCs. (**A**) GC1 and GC2 cells were treated with pemigatinib (500 nM) for 48 h, and *FGFR1*, *FOXM1*, and *S100A4* gene expression was assessed by qPCR, normalized to *GAPDH*. (**B**) S100A4 protein levels in GC1 and GC2 cells after pemigatinib treatment (500 nM, 48 h) were analyzed using flow cytometry. (**C**) *S100A4* gene expression was measured by qPCR in GC1 and GC2 cells transfected with *S100A4*-targeting siRNAs (si*S100A4*(1), si*S100A4*(8)) compared to scramble control (siCtl) and using *GAPDH* expression for normalization. (**D**) S100A4 protein levels in GC1 and GC2 cells after transfection with control siRNA (siCtrl) or two independent siRNAs targeting *S100A4* (si*S100A4*(1) and si*S100A4*(8)) were analyzed using flow cytometry. (**E**) Neurosphere formation of GC1 and GC2 cells transfected with si*S100A4*(1) and a scramble control (siscr) for 48 h following 2 to 6 Gy irradiation. Statistical significance was determined relative to control using Student’s *t*-test: * *p* ≤ 0.05, ** *p* < 0.01, *** *p* < 0.001, **** *p* < 0.0001.

**Figure 6 cells-14-01427-f006:**
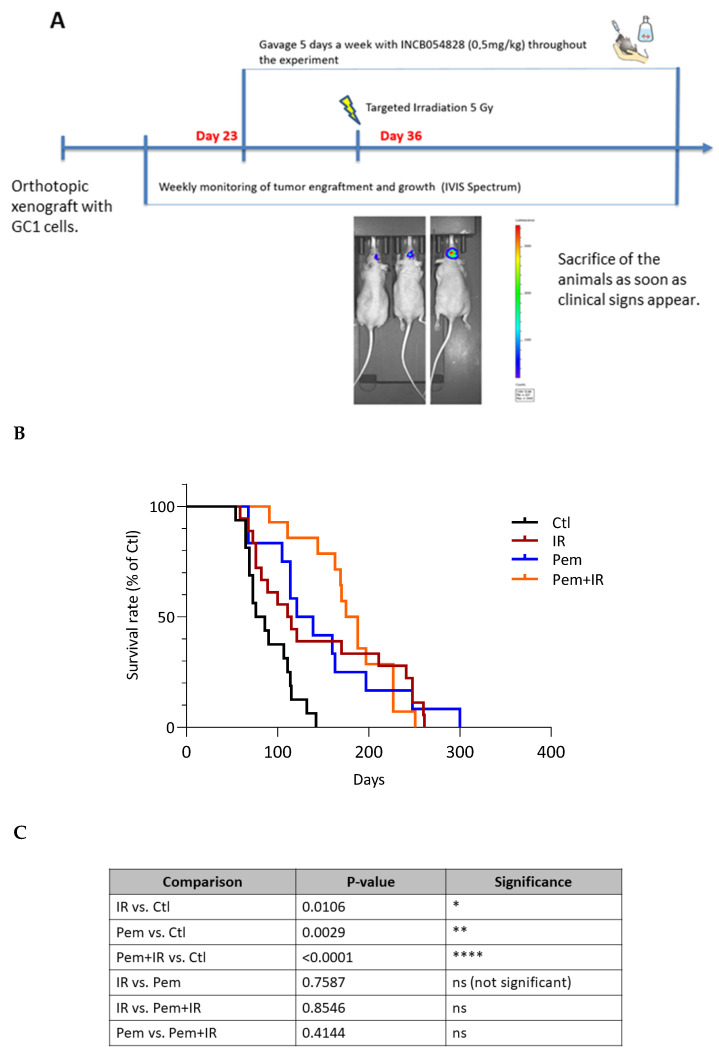
Kaplan–Meier survival analysis of mice treated with pemigatinib and/or irradiation. (**A**) Diagram of the experimental design. (**B**) Survival curves represent four experimental groups: untreated control (Ctl) (13 mice), irradiation alone (IR, 5 Gy) (17 mice), pemigatinib alone (Pem, 0.5 mg/kg) (13 mice), and combined treatment (IR + Pem) (14 mice). Mice were monitored daily, and survival was recorded over time. Data were plotted using the Kaplan–Meier method. (**C**) Statistical comparison of survival curves between treatment groups using Log-Rank Test *p*-values. Statistical significance: * *p* ≤ 0.05, ** *p* < 0.01, **** *p* < 0.0001.

## Data Availability

All the RNA-seq datasets generated in this study have been deposited in the NCBI Sequence Read Archive (SRA) under accession number PRJNA1320628.

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
