# Peer review of "FGFR1 Inhibition by Pemigatinib Enhances Radiosensitivity in Glioblastoma Stem Cells Through S100A4 Downregulation"

_cells, 2025, doi:10.3390/cells14181427_

Round 1
Reviewer 1 Report
Comments and Suggestions for Authors
The article is relevant for the study of the potential use of Pemigatinib for the treatment of glioblastoma, whose prognosis remains unfavorable for patients.
Convincing evidence in vitro is provided that combination pemigatinib with temozolomide, further enhanced radiosensitivity in unmethylated differentiated GBM cell lines.
Some exaggerations should be removed from the text before publication or additional experiments should be done
- It is impossible to state unequivocally about the transferability of the results to clinical practice, since only a limited number of cell models were studied, in which the main effects of this study were demonstrated. Are there any additional studies using primary glioblastoma?
- Secondly, there are no cytometry studies investigating the role of stem cell subset markers in radioresistance after pemigatinib treatment. This observation calls into question the validity of the statement that pemigatinib can serve as a valuable adjunct to radiotherapy, this statement should be softened. Could you provide any additional flow cytometry data about elimination of cancer stem cells after treatment of GBM with pemigatinib and radiotherapy?
- In vivo data contradict the hypothesis that pemigatinib can serve as a valuable adjunct to radiotherapy for newly diagnosed GBM, since the combination of pemigatinib and radiation did not improve survival compared with monotherapy. It is recommended to remove this exaggeration from the text.
Author Response
Comments 1: It is impossible to state unequivocally about the transferability of the results to clinical practice, since only a limited number of cell models were studied, in which the main effects of this study were demonstrated. Are there any additional studies using primary glioblastoma?
|
Response 1: We thank the reviewer for this thoughtful comment. We fully agree that preclinical findings must be interpreted with caution regarding their clinical transferability. However, we would like to emphasize that our study was conducted using both established GBM cell lines and two patient-derived glioblastoma stem cell (GSC) lines, GC1 and GC2, which better reflect the complexity and heterogeneity of primary tumors. Importantly, pemigatinib—the FGFR1-3 inhibitor investigated here—is already approved for clinical use in patients with FGFR2-altered cholangiocarcinoma, with ongoing clinical trials exploring its activity in glioblastoma (e.g., NCT05267106). This existing clinical approval provides well-established pharmacokinetic, safety, and dosing data, reducing the translational barrier for repurposing in GBM. We added to the Discussion section accordingly to reflect this point more clearly. Discussion (Section 4, Page 20, lane 745) “Further is the use of a limited number of primary GSC models.. While GC1 and GC2 reflect clinically relevant MGMT-unmethylated GBM subtypes, additional studies across a broader panel of primary GSCs and patient-derived xenografts are needed to confirm the generalizability of these findings and to better define the subset of GBM patients most likely to benefit from FGFR1-targeted radiosensitization.”
|
Comments 2: Secondly, there are no cytometry studies investigating the role of stem cell subset markers in radioresistance after pemigatinib treatment. This observation calls into question the validity of the statement that pemigatinib can serve as a valuable adjunct to radiotherapy, this statement should be softened. Could you provide any additional flow cytometry data about elimination of cancer stem cells after treatment of GBM with pemigatinib and radiotherapy? |
Response 2: We thank the reviewer for this thoughtful comment. In our study, we employed two well-characterized patient-derived glioblastoma stem cell (GSC) lines (GC1 and GC2), previously validated in both phenotypic and functional assays as bona fide GSCs (Gouazé-Andersson et al., Oncotarget 2018; Lemarié et al., Sci Adv 2023). These models exhibit hallmark GSC features, including self-renewal capacity, expression of stemness-associated genes, and resistance to standard therapies. We assessed the impact of pemigatinib ± radiation using gold-standard functional assays for GSC biology: sphere formation, clonogenic survival, and neurosphere size measurements. These readouts directly reflect the self-renewal and tumor-propagating potential of GSCs, and our results consistently demonstrated a dose-dependent impairment of these properties following treatment. While flow cytometry–based analysis of stem cell surface or intracellular markers (e.g., Olig2, Sox2, Nestin) could provide complementary phenotypic information, such markers are heterogeneously expressed among GSC subsets and are not always definitive indicators of functional stemness. For this reason, we prioritized functional assays to capture the biological consequences of treatment on the stem-like compartment. We nevertheless agree that phenotypic characterization could strengthen the interpretation and have modified the manuscript to acknowledge this limitation and outline future plans to incorporate such analyses. Changes in the manuscript: We have revised the Abstract and Discussion as follows: "" Abstract (final sentence)(page 1, lane 33) Discussion (Section 4, page 20, lane 754):
Comments 3: In vivo data contradict the hypothesis that pemigatinib can serve as a valuable adjunct to radiotherapy for newly diagnosed GBM, since the combination of pemigatinib and radiation did not improve survival compared with monotherapy. It is recommended to remove this exaggeration from the text.
Response 3: We appreciate the reviewer’s thoughtful comment. It is correct that, in our in vivo model using GC1 cells, the combination of pemigatinib and irradiation did not result in a statistically significant improvement in survival compared to monotherapy arms. However, we respectfully note that the combination treatment yielded the longest median overall survival (177.7 days), exceeding both radiotherapy alone (118.5 days) and pemigatinib alone (140.7 days). While the differences did not reach statistical significance, this trend toward extended survival suggests potential clinical benefit in combining both treatments — particularly in the context of MGMT-unmethylated glioblastoma, which is notoriously treatment-resistant. To reflect the reviewer’s comment, we have softened the original statement in the manuscript, and now present the in vivo findings with more nuance, acknowledging the absence of statistical significance while retaining the potential translational interest of the combination. Abstract (page 1, lane 33): Discussion (Section 4, page 20, lane 770): Conclusion (Section 5, page 21, lane 793):
|

Reviewer 2 Report
Comments and Suggestions for Authors
The manuscript by Gouazé-Andersson aims to analyze the use of pemigatinib in the radiosensitization of GBM cells. Multiple methodological strategies were employed, ranging from bioinformatics to xenografts. However, I identified several caveats, including: i) an unclear rationale connecting sequential experiments; ii) lack of proper controls and/or incomplete analysis; iii) incomplete description of methods; and iv) significant data interpretation issues.
Major observations
- The abstract needs rewording to improve clarity and precision.
- Please include proper citations for all statements regarding previously published data. For example, lines 54–57 contain three statements about the role of FGFR1 and HIF1α in resistance, but no references are provided.
- Some information is oversimplified. For example, the authors highlight the importance of integrin alpha6 in FGFR1 signaling (lines 80–84). However, FGFR1 is not functionally related to inhibition of integrins αvβ3 and αvβ5. The statement "The crosstalk between FGFR1 and integrins might explain the unsuccessful outcome of the CENTRIC trial, which utilized a selective αvβ3 and αvβ5 integrin inhibitor" is misleading. These integrins are encoded by different genes and have distinct functions compared to α6-containing integrins.
- The introduction lacks scientific detail. For example, the phrase "What sets pemigatinib apart from earlier FGFR inhibitors is its heightened selectivity for members of the FGFR family" requires supporting evidence. What are the IC50 values for FGFRs? What other kinases have been tested, and what is the selectivity index?
- In section 2.1, the authors state, "The characteristics of GC1 and GC2 as stem cells have been previously detailed by our group [11]." Please provide evidence that the key characteristics reported in 2018 are still present in the cultures used in this study. This is essential given the focus on stem cell behavior.
- In reference [11], GC1 and GC2 were studied in both neurosphere (stem) and differentiated conditions. What is the rationale for using the unrelated U87 and LN18 lines instead of differentiated GC1 and GC2? Wouldn't the latter provide a better comparison model for the experimental question?
- Section 2.9 describes RNA extraction and sequencing but lacks details about conditions such as cell density, drug concentration, exposure time, number of replicates, and RNA-seq analysis workflow. As a result, the method used to generate the data shown in Figure 4 is unclear. Also, why is there only one transcriptomic analysis covering both cell models?
- What was the rationale behind using 1–3 μM concentrations of pemigatinib in some experiments and 125–500 nM in others? Why was 1 μM chosen for combination therapy experiments (Figure 2)?
- The labeling of Figures 1 and 4C is unclear.
- The evaluation of phospho-FRS2 (Figures 2A and 3A) is invalid because: a) it must be normalized to total FRS2; b) there is no quantitative data or statistical comparison (despite claiming triplicate experiments); c) when attempting normalization with β-actin, the blots appear overexposed, compromising linearity. These experiments must be repeated.
- Blots in Figure 2A differ significantly in contrast from the original ones. Were they digitally edited?
- I could only locate the original blots for actin from Figure 2A. There are no identifiable blots for 3A, likely due to improper labeling. If they cannot be tracked to a figure, their value is null.
- Sphere size (Figure 3F) is not a valid measurement of clonogenicity, as it is influenced by multiple processes (e.g., survival and proliferation).
- Figure S1 lacks statistical comparisons and only includes data for GC2. Therefore, it is incorrect to state: "Pemigatinib significantly impaired sphere formation in both lines and enhanced the effects of irradiation, confirming its potential in TMZ-refractory GSCs." (Section 3.5)
- What was the rationale for selecting a few genes from the >1,400 differentially expressed? If gene selection was arbitrary or literature-based, performing RNA-seq seems unjustified.
- Figure 5B shows that pemigatinib reduced expression by 10–15%. What is the biological (not just statistical) significance of such a modest reduction?
- Include measurements of S100A4 protein expression after gene silencing.
- Please present tumor growth kinetics as measured by bioluminescence imaging (BLI), since the authors used luc+ cells and claimed to have appropriate equipment.
- In vivo experiments show that the combination of pemigatinib and radiotherapy improves mouse survival. This is a key finding, but: i) mechanistic analysis is lacking — at least two selected target genes should be evaluated in tumors; ii) survival analysis alone is insufficient to demonstrate changes in the CSC pool — conclusions are not supported by the data presented.
- Figure 6A and Section 2.12 do not match in their methodological description.
- The discussion omits important findings: i) FGFR3 is overexpressed in normal tissue compared to GBM. Since pemigatinib targets FGFR3, what would be its effect on healthy tissue? ii) Tumor resistance to pemigatinib has been reported. Could this occur in GBM patients?
Minor observations
22. There are multiple typos, including critical ones (e.g., "Mice received pemigatinib (0. mg/kg, oral gavage)" or incorrect use of FoxM1 instead of FOXM1).
23. Some figures show digital editing in the embedded text.
24. Avoid non-quantitative terms such as "lower doses" or "clear radiosensitization". Provide numerical values and refer to the corresponding results or figures.
25. Use SI units and abbreviations consistently throughout the manuscript.
Author Response
Comments 1. The abstract needs rewording to improve clarity and precision.
Response 1. We thank the reviewer for this suggestion. The Abstract has been fully revised to improve clarity, precision, and logical flow.
Comments 2. Please include proper citations for all statements regarding previously published data. For example, lines 54–57 contain three statements about the role of FGFR1 and HIF1α in resistance, but no references are provided.
Response 2. We thank the reviewer for pointing out this omission. We have now added appropriate references to support each of the statements concerning the role of FGFR1 and HIF1α in radioresistance. Specifically:
- Line 54–57, original text:
“In differentiated GBM cells, combining FGFR1 inhibition with radiation leads to centrosome overduplication, mitotic cell death, and decreased HIF1α expression. While HIF1α plays a key role in hypoxic responses, it also independently contributes to radioresistance. FGFR1 inhibition similarly delays the growth of irradiated tumor xenografts, a process linked to reduced HIF1α levels without affecting blood vessel integrity.”
Revision with citations added:
Introduction (Section 1, page 2, lane 54):
“In differentiated GBM cells, combining FGFR1 inhibition with radiation has been shown to induce centrosome overduplication, mitotic catastrophe, and a decrease in HIF1α levels, thereby enhancing radiosensitivity [Gouazé-Andersson, 2016]. While HIF1α is a key mediator of hypoxic adaptation, it also independently promotes radioresistance through transcriptional regulation of survival pathways [Marampon, 2014; Song, 2025]. In vivo, FGFR1 inhibition delayed the growth of irradiated xenografts, an effect linked to reduced HIF1α expression, without altering vascular density [Gouazé-Andersson, 2016].”
Comments 3. Some information is oversimplified. For example, the authors highlight the importance of integrin alpha6 in FGFR1 signaling (lines 80–84). However, FGFR1 is not functionally related to inhibition of integrins αvβ3 and αvβ5. The statement "The crosstalk between FGFR1 and integrins might explain the unsuccessful outcome of the CENTRIC trial, which utilized a selective αvβ3 and αvβ5 integrin inhibitor" is misleading. These integrins are encoded by different genes and have distinct functions compared to α6-containing integrins.
Response 3. We thank the reviewer for this accurate and helpful observation. We agree that the statement linking the FGFR1–α6-integrin crosstalk to the failure of the CENTRIC trial is speculative and biologically unfounded, as αvβ3/αvβ5 integrins targeted in that trial are distinct in function and gene origin from α6-containing integrins. To avoid any confusion or overinterpretation, we have removed the reference to the CENTRIC trial from this paragraph.
To address this and clarify the mechanistic context, we have revised the paragraph to focus specifically on the validated interaction between integrin α6 and FGFR1 in glioblastoma stem cells:
Revised sentence in the manuscript
Introduction (Section 1, page 2, lane 80):
“Furthermore, an interaction between FGFR1 and integrin α6 has been identified, demon-strating that α6-integrin contributes to GSC proliferation and stemness by regulating FGFR1 and FOXM1 expression via the ZEB1/YAP1 transcriptional complex [16]. This α6-integrin–FGFR1 crosstalk underscores the importance of the FGFR1 axis in sustaining glioblastoma stem cell properties and supports its relevance as a therapeutic target in GBM.”
Comments 4. The introduction lacks scientific detail. For example, the phrase 'What sets pemigatinib apart from earlier FGFR inhibitors is its heightened selectivity for members of the FGFR family' requires supporting evidence. What are the IC50 values for FGFRs? What other kinases have been tested, and what is the selectivity index?
Response 4. We thank the reviewer for highlighting the need for more scientific specificity regarding pemigatinib’s selectivity. We have revised the Introduction to include quantitative data on IC₅₀ values for FGFR1–3, as well as its kinase selectivity profile across a broader panel.
Specifically, pemigatinib (INCB054828) has demonstrated low nanomolar potency against FGFR1 (IC₅₀ ≈ 0.4 nM), FGFR2 (0.5 nM), and FGFR3 (1.0 nM), and exhibits over 100-fold selectivity relative to most other kinases tested in a panel of more than 100 kinases [Ref: Liu et al., PLoS One, 2020]. We have added this information along with a citation in the revised manuscript.
Although pemigatinib shows IC₅₀ values in the low nanomolar range for FGFR1–3 in biochemical assays, the concentrations used in our study (≥125 nM) were necessary to achieve effective pathway inhibition and functional effects in glioblastoma stem cells (GSCs), which are known to display intrinsic resistance and reduced drug permeability. We have added this clarification to the manuscript to better contextualize our dosing strategy.
Revised Sentence for the Manuscript
Introduction (Section 1, page 2, lane 85)
“Pemigatinib, also known as INCB054828, distinguishes itself from earlier FGFR inhibitors through its high potency and selectivity for FGFR1–3, with reported IC₅₀ values of approximately 0.4 nM for FGFR1, 0.5 nM for FGFR2, and 1.0 nM for FGFR3, while showing markedly weaker activity against FGFR4 (IC₅₀ ≈ 30 nM) [. In a broad kinase selectivity screen including more than 100 kinases, pemigatinib exhibited over 100-fold selectivity for FGFR1–3 relative to the vast majority of other targets, with only a few non-FGFR kinases with IC₅₀ values below 1,000 nM, including VEGFR-2 (KDR; IC₅₀ ≈ 190 nM) and c-Kit (IC₅₀ ≈ 270 nM) [17]. These values highlight its ultra-low nanomolar biochemical potency and strong target specificity. In our study, however, higher concentrations were required to elicit functional effects in glioblastoma stem cells, likely reflecting the intrinsic resistance mechanisms and reduced drug permeability characteristic of GSCs compared to conventional cancer cell lines.”
Comments 5. In section 2.1, the authors state, 'The characteristics of GC1 and GC2 as stem cells have been previously detailed by our group [11].' Please provide evidence that the key characteristics reported in 2018 are still present in the cultures used in this study. This is essential given the focus on stem cell behavior.
Response 5. We thank the reviewer for raising this important concern. GC1 and GC2 were both thoroughly characterized in our prior study [11] for their expression of stem cell markers, self-renewal capacity, and tumorigenicity. Following initial characterization, early-passage stocks (passage 2 or 3) were cryopreserved. For the current study, experiments were performed using cells cultured up to a maximum of passage 10. This controlled propagation strategy helps preserve the original phenotypic properties and minimizes the risk of cellular drift.
We have revised Section 2.1 of the manuscript to clarify this:
Materials and Methods (Section 2.1, page 3, lane 134)
“GC1 and GC2 were previously characterized as glioblastoma stem cell lines [13] and cryopreserved at early passages (P2–P3). To preserve their stem-like characteristics, all experiments in this study were conducted using cells between passages 3 and 10 after initial derivation from patient tumor tissue.
Comments 6. “In reference [11], GC1 and GC2 were studied in both neurosphere (stem) and differentiated conditions. What is the rationale for using the unrelated U87 and LN18 lines instead of differentiated GC1 and GC2? Wouldn't the latter provide a better comparison model for the experimental question?”
Response 6. We thank the reviewer for this thoughtful question regarding our model choice. Our rationale for including U87 and LN18 cells was based on the need to assess the effect of pemigatinib in differentiated glioblastoma cells with differing MGMT methylation status, which is central to the response to temozolomide (TMZ). Specifically, LN18 cells are MGMT-unmethylated (resistant to TMZ), while U87 cells are MGMT-methylated (sensitive to TMZ). In contrast, both GC1 and GC2 stem lines are MGMT-unmethylated. This design allowed us to better explore the relationship between MGMT status, pemigatinib treatment, and TMZ sensitivity in differentiated GBM cells.
While differentiated GC1 and GC2 cells could theoretically serve as more direct comparisons, preliminary experiments showed that FGFR1 expression was not significantly modulated by pemigatinib at concentrations effective in the stem state (125-500 nM). These results are now included as Supplementary Figure S2, which shows no detectable modulation of FGFR1 protein levels by pemigatinib in differentiated GC1 or GC2 cells, suggesting reduced FGFR1 pathway activity in this state or less accessible in the differentiated state.
We added this sentence to the discussion and the supplemental figure Fig S2.
Discussion (Section 4, page 18, lane 636)
“While differentiated GC1 and GC2 cells could theoretically serve as more direct comparators, our preliminary experiments showed that FGFR1 expression was not significantly altered by pemigatinib at concentrations effective in the stem cell state (125-500 nM). As shown in Supplementary Figure S2, FGFR1 protein levels remained unchanged in differentiated GC1 and GC2 cells, suggesting that FGFR1 pathway activity is reduced or less accessible in the differentiated state.”
Supplemental Figure Fig S2 (Appendix C, page 24, lane 856)
Legend Fig S2. (Appendix C, page 24, lane 860)
“Figure S2. Western blot analysis of FGFR1 expression in differentiated glioblastoma stem cells. GC1 and GC2 were differentiated for two weeks before treatment with increasing concentrations of pemigatinib (125, 250, 500 nM) for 48 h. Untreated cells (0 nM) served as controls. β-actin was used as a loading control.”
Comments 7. “Section 2.9 describes RNA extraction and sequencing but lacks details about conditions such as cell density, drug concentration, exposure time, number of replicates, and RNA-seq analysis workflow. As a result, the method used to generate the data shown in Figure 4 is unclear. Also, why is there only one transcriptomic analysis covering both cell models?”
Response 7. We thank the reviewer for this valuable comment. In our study, RNA sequencing was performed on both GC1 and GC2 glioblastoma stem cell lines, each treated with pemigatinib or vehicle control. The transcriptomic results presented in Figure 4 reflect the integrated analysis of differential gene expression across both models, with results averaged across the two datasets. We recognize that this was not clearly explained and have revised Section 2.9 to include methodological details.
Revised Sentence for the Manuscript
Materials and Methods (Section 2.9, page 5, lane 204)
“For transcriptomic analysis, GC1 and GC2 glioblastoma stem cell lines were seeded at 2 × 10⁶ cells in 5 mL medium in 25 cm³ flasks and treated with either 250 nM pemigatinib or DMSO control for 48 h (three independent biological replicates per condition per cell line, n = 6 total per condition). RNA was extracted using the RNeasy kit (Qiagen) and quality verified (RQN > 8) prior to library preparation with the Illumina Stranded Total RNA Prep protocol, including rRNA depletion, strand-specific library construction, and paired-end sequencing. Library quality and concentration were assessed using Qubit™ dsDNA BR Assay (Thermo Fisher Scientific), High Sensitivity NGS Fragment Analysis Kit (Agilent Technologies), and KAPA Library Quantification Kit for Illumina® (Roche). Sequencing was performed on an Illumina NextSeq 550 with a High Output flow cell, and raw reads were demultiplexed with bcl2fastq v2.20.0.422. Transcriptomic results shown in Figure 4 represent an integrated differential expression analysis across both models, averaged to highlight changes conserved between GSC lines.”
Figure 4 Legend (page 14, lane 508)
“Figure 4. Analysis of genes and pathways affected by pemigatinib treatment. (A) Differential gene expression between glioblastoma stem cells (GC1 and GC2) treated with pemigatinib (250 nM, 48 h) and corresponding vehicle controls, based on an integrated analysis of both models (n = 3 biological replicates per condition per cell line). Data are presented as a volcano plot. (B) Gene ontology analysis identified pathways downregulated by pemigatinib treatment. (C) Box plot comparing S100A4 expression in glioblastoma (GBM) tumor tissue (T, n = 163) versus normal brain (N, n = 207) using TCGA/GTEx RNA-seq datasets.”
Comments 8. What was the rationale behind using 1–3 μM concentrations of pemigatinib in some experiments and 125–500 nM in others? Why was 1 μM chosen for combination therapy experiments (Figure 2)?
Response 8. We thank the reviewer for this important question. The concentration range of 125–500 nM was used in glioblastoma stem cell (GSC) experiments because these doses produced marked biological effects in stem-like cultures. In contrast, differentiated GBM cells (U87 and LN18) were less sensitive, and measurable effects were only observed at higher concentrations (≥ 1 μM). For the combination experiments with temozolomide (Figure 2), we selected 1 μM as it was the lowest concentration to elicit a detectable effect in differentiated cells, allowing us to evaluate whether minimal FGFR1 inhibition could influence TMZ response, particularly in MGMT-unmethylated cells, while minimizing the risk of off-target effects. This rationale has been added to the revised Methods and Discussion sections.
Materials and Methods (Section 2.3, page 4, lane 149)
“For combination experiments with temozolomide (100 mM) in differentiated GBM cells, 1 μM pemigatinib was chosen as the lowest concentration that produced a detectable biological effect in these cells, enabling assessment of potential interaction with TMZ while minimizing off-target effects.”
Discussion (Section 4, page 18, lane 624)
“Although we did not formally assess drug synergy, using 1 μM pemigatinib, the lowest active concentration in differentiated cells, enabled us to test whether modest FGFR1 inhibition could influence TMZ response, particularly in MGMT-unmethylated models, suggesting therapeutic potential in patients with limited response to alkylating agents.”
Comments 9. The labeling of Figures 1 and 4C is unclear.
Response 9. We thank the reviewer for pointing this out. We have revised the figure legends for Figures 1 and 4C to explicitly define “T” as glioblastoma tumor samples (n = 163) and “N” as normal brain tissue (n = 207), and to specify that values are expressed as log₂(TPM + 1) from TCGA/GTEx RNA-seq datasets. In addition, we have clarified in the Materials and Methods section that TPM refers to “transcripts per million.” These changes should improve clarity and facilitate interpretation of the figures.
Legend figure 1 (page 8, lane 307)
“Figure 1. Expression levels of FGFR1–FGFR4 in glioblastoma (GBM) tumor tissue (T, n = 163) and normal brain tissue (N, n = 207), based on TCGA/GTEx RNA-seq datasets accessed via the GEPIA web server. Data are shown as log₂(TPM + 1) values. Statistical comparisons were performed using one-way ANOVA; 0.01<p< 0.05 is indicated by an asterisk.”
Legend figure 4C (page 14, lane 512)
“Figure 4C. Box plot comparing S100A4 expression in glioblastoma (GBM) tumor tissue (T, n = 163) versus normal brain tissue (N, n = 207) using TCGA/GTEx RNA-seq datasets via the GEPIA web server. Data are shown as log₂(TPM + 1) values. Statistical analysis was performed using one-way ANOVA; p< 0.05 is indicated by an asterisk.”
Materials and Methods (Section 2.11, page 7, lane 262)
“Gene expression values are presented as log₂(TPM + 1), where TPM denotes transcripts per million.”
Comments 10. The evaluation of phospho-FRS2 (Figures 2A and 3A) is invalid because: a) it must be normalized to total FRS2; b) there is no quantitative data or statistical comparison (despite claiming triplicate experiments); c) when attempting normalization with β-actin, the blots appear overexposed, compromising linearity. These experiments must be repeated.
Response 10. We appreciate this rigorous feedback. We have now probed and displayed total FRS2 together with phospho-FRS2 (p-FRS2) on the same membranes in the revised Figures 2A and 3A. Total FRS2 expression does not change across conditions, indicating that the p-FRS2 signal reflects phosphorylation status rather than protein abundance. The blots are not overexposed; exposures were set within the linear detection range.
Comments 11. Blots in Figure 2A differ significantly in contrast from the original ones. Were they digitally edited?
Response 11. We thank the reviewer for pointing this out. To dispel any concern about the Western blot, we re-exported the panel directly from the raw chemiluminescence files, probing p-FRS2 and total FRS2 on the same membranes and ensuring exposures within the linear detection range. For figure assembly, only uniform, global brightness/contrast adjustments were applied to the entire image—no band-specific or local edits, no splicing, and no aspect-ratio changes. The uncropped, unprocessed originals (with exposure series) are provided in the separate document.
Comments 12. I could only locate the original blots for actin from Figure 2A. There are no identifiable blots for 3A, likely due to improper labeling. If they cannot be tracked to a figure, their value is null.
Response 12. We thank the reviewer for pointing this out. Uncropped blots were inadvertently omitted/mislabeled in the first version. The revision rectifies this: every Western blot is now traceable and properly labeled, and all uncropped, unprocessed blots for Fig. 2A and Fig. 3A (p-FRS2, total FRS2, FGFR1, and β-actin) are provided.
Comments 13. Sphere size (Figure 3F) is not a valid measurement of clonogenicity, as it is influenced by multiple processes (e.g., survival and proliferation).
Response 13. We thank the reviewer for this observation. As shown in Figure 3F, pemigatinib significantly reduced the size of neurospheres formed by GC1 and GC2 glioblastoma stem cells under both basal (0 Gy) and irradiated (4 Gy) conditions. While we agree that sphere size is not a direct measure of clonogenicity, it provides complementary information by reflecting the proliferative capacity of cells within each sphere after treatment. This parameter, together with our clonogenic survival data (Figure 3E), offers a more complete picture of the impact of pemigatinib on GSC growth and radiosensitivity.
Legend figure 3F. (page 12, lane 446)
(F). GC1 and GC2 cells were subjected to a clonogenic assay and treated with pemigatinib (250 nM or 500 nM), with or without 4 Gy irradiation. At the end of the assay (day 10), sphere size (in μm) was measured using Fiji software as an indicator of the proliferative capacity of the cells within each sphere.
Results (Section 3.4, page 11, lane 417)
“In this context, sphere size at the end of the clonogenic assay was used as an indicator of the proliferative capacity of cells within each sphere, complementing the survival fraction data.”
Comments 14. Figure S1 lacks statistical comparisons and only includes data for GC2. Therefore, it is incorrect to state: "Pemigatinib significantly impaired sphere formation in both lines and enhanced the effects of irradiation, confirming its potential in TMZ-refractory GSCs." (Section 3.5)
Response 14. We thank the reviewer for this observation. Figure S1 was intended as supplemental and descriptive data illustrating the effect of pemigatinib in GC2 cells under various irradiation doses, rather than a statistically analyzed dataset. We agree that, as presented, these data should not be used to support a statement of statistical significance or a generalization to both GSC lines.
Accordingly, we have revised the text in Section 3.5 to read:
Results (Section 3.5, page 12, lane 457)
“In GC2 cells, pemigatinib alone or in combination with TMZ appeared to reduce sphere formation, with this trend being maintained under increasing doses of irradiation. These supplemental data illustrate the potential effect of pemigatinib in TMZ-refractory GSCs and warrant further validation.”
Legend FigS1 (Appendix A, page 23, lane 840)
Figure S1. Effect of pemigatinib in association with standard treatments on GC2 glioblastoma stem-like cells. Sphere formation was assessed by clonogenic assay after GC2 cells were treated with pemigatinib (125 nM), temozolomide (100 μM), and irradiation (2–6 Gy). Bar graphs represent the mean ± SD of sphere numbers per 100 cells from descriptive replicate experiments in GC2 cells only; no statistical analysis was performed. Photomicrographs correspond to representative clonogenic assays at 4 Gy, illustrating the observed trends.
Comments 15. What was the rationale for selecting a few genes from the >1,400 differentially expressed? If gene selection was arbitrary or literature-based, performing RNA-seq seems unjustified.
Response 15. We thank the reviewer for this important comment. We would like to emphasize that the RNA-seq analysis conducted in this study was not performed solely to support the current manuscript, but rather as part of a broader research strategy aimed at characterizing the molecular consequences of FGFR1 inhibition in glioblastoma stem cells (GSCs). The dataset is currently being utilized in multiple ongoing projects beyond the scope of this paper.
In this specific study, we focused on a limited number of genes—such as FGFR1, FOXM1, and S100A4—based on:
(i) their well-established roles in GSC biology, mesenchymal transition, and radioresistance;
(ii) their consistent and significant downregulation in our RNA-seq dataset; and
(iii) prior mechanistic data from our group showing regulation of these targets by FGFR1 using siRNA.
These genes were therefore not selected arbitrarily or solely based on literature, but rather through a combination of biological relevance, transcriptomic results, and prior functional evidence.
Importantly, we validated their involvement using independent functional approaches (qPCR, flow cytometry, siRNA knockdown, and clonogenic assays), confirming particularly the contribution of S100A4 to radiosensitivity.
Thus, the RNA-seq served both as a hypothesis-generating tool for this study and as a foundational dataset for future investigations. We believe this integrated approach strengthens the biological significance and reproducibility of our findings.
Comments 16. Figure 5B shows that pemigatinib reduced expression by 10–15%. What is the biological (not just statistical) significance of such a modest reduction?
Response 16. We thank the reviewer for this insightful question. While the reduction in S100A4 protein levels observed by flow cytometry in Figure 5B appears modest (approximately 10–15%), we believe it is biologically meaningful for several reasons:
- Amplified downstream effects: S100A4 is a multifunctional protein that acts as a regulator of several signaling cascades associated with mesenchymal transition, migration, and therapy resistance. Even modest decreases in its expression can disrupt these finely tuned signaling networks, especially in glioblastoma stem cells where signaling thresholds are critical.
- Complementary transcriptional and functional data: Although the protein-level reduction measured by flow cytometry is modest, qPCR data (Figure 5A) showed a more pronounced downregulation of S100A4 mRNA. More importantly, siRNA-mediated knockdown of S100A4 (Figure 5C-D) significantly enhanced radiosensitivity, directly demonstrating its functional relevance. This supports the idea that even partial downregulation (as seen with pemigatinib) contributes to radiosensitization.
- Context of combinatorial treatment: The modest reduction in S100A4 expression by pemigatinib occurs alongside multiple other transcriptomic and functional changes, including downregulation of FOXM1 and cell cycle/DNA repair pathways (Figure 4). Therefore, this reduction contributes to radiosensitization in a cumulative and synergistic manner, rather than acting alone.
- Detection sensitivity limitations: Flow cytometry may underestimate protein reduction, especially for intracellular proteins like S100A4 that are not uniformly accessible or stable. The biological effect may be greater than what is captured by surface or cytoplasmic fluorescence intensity alone.
Comments 17. Include measurements of S100A4 protein expression after gene silencing.
Response 17. We thank the reviewer for this valuable suggestion. In response, we evaluated S100A4 protein expression following siRNA-mediated silencing in GC1 and GC2 glioblastoma stem cells by flow cytometry. The analysis showed a reduction of 37–49% in GC1 and 55% in GC2 after transfection with two independent siRNAs compared with the control siRNA (siCtrl), consistent with our mRNA expression results. These data have been added to the revised manuscript (new Figure 5D).
RESULTS (Section 3.8, page 14, lane 527)
“Flow cytometry analysis confirmed efficient S100A4 silencing at the protein level, showing a 37–49% reduction in GC1 and a 55% reduction in GC2 compared with siCtrl, in line with the mRNA expression data (Fig 5D).”
Fig 5D (Page 15, lane 537)
Insertion Fig5D
Legend Fig 5D (page 16, lane 553)
“(D) S100A4 protein levels in GC1 and GC2 cells after transfection with control siRNA (siCtrl) or two independent siRNAs targeting S100A4 (siS100A4(1) and siS100A4(8)) were analyzed using flow cytometry.”
Comments 18. Please present tumor growth kinetics as measured by bioluminescence imaging (BLI), since the authors used luc+ cells and claimed to have appropriate equipment.
Response 18. We thank the reviewer for this suggestion. We did not use BLI to monitor tumor growth kinetics during treatment, as anatomical variability in tumor depth and tumor infiltration can influence photon emission independently of tumor size, thereby limiting the reliability of longitudinal measurements. However, we did use BLI to confirm tumor implantation and progression before treatment initiation. For the reviewer’s consideration, we provide below representative BLI images and quantification from a cohort of mice implanted with luc⁺ GSCs, which confirm successful engraftment and tumor growth. These data were used solely for implantation control and are not included in the revised manuscript, since BLI was not used to evaluate the effect of pemigatinib on tumor growth.
Figure provided for reviewer only. Representative BLI images and quantification of photon flux in mice implanted with luc⁺ GSCs. These data confirm successful tumor implantation and growth prior to treatment initiation. BLI was not used for longitudinal monitoring of tumor growth or to assess the effect of pemigatinib.
Comments 19. In vivo experiments show that the combination of pemigatinib and radiotherapy improves mouse survival. This is a key finding, but: (i) mechanistic analysis is lacking — at least two selected target genes should be evaluated in tumors; (ii) survival analysis alone is insufficient to demonstrate changes in the CSC pool — conclusions are not supported by the data presented.Response 19. We thank the reviewer for this valuable comment. The primary aim of our in vivo experiments was to determine the therapeutic efficacy of pemigatinib in combination with radiotherapy using a clinically relevant GSC-derived orthotopic GBM model. For this objective, survival analysis represents the most direct and appropriate endpoint. While mechanistic analyses in vivo (e.g., target gene expression in tumors) could provide additional insights, they are not required to establish therapeutic efficacy. Unfortunately, such analyses have not be performed. We have therefore relied on our in vitro mechanistic data to support the involvement of S100A4 and FOXM1 pathways, and we now explicitly acknowledge this limitation in the Discussion.
Regarding the CSC population, We did not intend to investigate changes in the CSC pool in vivo, and the manuscript does not make such claims. To avoid any possible misunderstanding, we have revised the Discussion to state explicitly that the survival benefit observed in our orthotopic model is interpreted solely as evidence of therapeutic efficacy, without inference on CSC dynamics.
The title of the section 3.9 have been modified.
Results (Section 3.9, page 16, lane 558)
“3.9. In vivo study of the therapeutic efficacy of pemigatinib on human GSCs orthotopic xenografts.”
Discussion (Section 4, page 20, lane 770)
“Although the combination did not statistically outperform either monotherapy, it achieved the longest median overall survival (177.7 days). This outcome, while not formally synergistic, is clinically meaningful, especially given the limited options for MGMT-unmethylated GBM. However, given the lack of statistical significance, we recognize that further studies are required to confirm the clinical relevance of this combined approach. The in vivo experiments were designed to assess therapeutic efficacy, and the observed survival benefit is interpreted in this context only, without inference on CSC dynamics.”
Comments 21. The discussion omits important findings: i) FGFR3 is overexpressed in normal tissue compared to GBM. Since pemigatinib targets FGFR3, what would be its effect on healthy tissue? ii) Tumor resistance to pemigatinib has been reported. Could this occur in GBM patients?
Response 21. We thank the reviewer for these important comments, which raise two key translational considerations. We have revised the Discussion accordingly to address both the potential impact of FGFR3 inhibition in normal brain tissue and the possibility of acquired resistance to pemigatinib in GBM.
1) FGFR3 expression in normal tissue and potential effects of pemigatinib
We agree that FGFR3 is more highly expressed in normal brain compared to GBM, as shown in our GEPIA analysis (Fig. 1). Since pemigatinib inhibits FGFR1–3, potential effects on healthy brain tissue cannot be excluded. However, two factors are likely to mitigate this concern: (1) pemigatinib shows strong selectivity for FGFR1–3 over non-FGFR kinases (>100-fold), thereby limiting broad off-target kinase inhibition; (2) although the BBB is frequently disrupted in GBM lesions, allowing drug penetration into tumors, it remains largely intact in healthy parenchyma, which may restrict pemigatinib access to FGFR3-expressing normal tissue. Importantly, clinical data from trials such as FIGHT-202 report mainly on-target adverse effects, including hyperphosphatemia due to altered phosphate handling, as well as mucosal toxicities, nail and skin changes, and ocular effects. These toxicities are considered manageable and largely reversible with dose adjustments, phosphate binders, and supportive care (Vogel A, ESMO Open 2024; Abou-Alfa GK, Lancet Oncol 2020). Neurological side effects have not been a prominent finding, supporting limited CNS toxicity in clinical use. We have now added this point to the Discussion.
2) Acquired resistance to pemigatinib
We acknowledge that resistance to pemigatinib has been reported in other cancers, most notably cholangiocarcinoma, due to secondary kinase domain mutations or bypass pathway activation. While these mechanisms have not yet been described in GBM, the inherent plasticity of glioblastoma stem cells and frequent activation of alternative growth factor receptors (EGFR, MET, PDGFR) suggest that resistance could occur. This possibility strengthens the rationale for evaluating pemigatinib in combination with radiotherapy or temozolomide, as investigated in our study, rather than as a monotherapy. We have now added a paragraph to the Discussion to highlight this translational consideration.
Discussion (Section 4, page 19, lane 676)
“Together, these results suggest that pemigatinib can exert radiosensitizing effects at least in part through suppression of stemness and mesenchymal programs driven by FGFR1 and its downstream effectors FOXM1 and S100A4. These mechanistic insights indicate that FGFR inhibition may complement existing therapeutic strategies in GBM, although further studies will be needed to confirm this in additional models. Importantly, this effect was observed despite the absence of FGFR gene fusions or known activating mutations in our models, suggesting that functional FGFR1 pathway activity, rather than mutational status alone, may contribute to sensitivity to FGFR inhibition. This raises the possibility of a shift in biomarker strategies, moving from purely genomic profiling toward assessing pathway dependency, thereby potentially broadening the scope of patients who could benefit from FGFR-targeted therapy.
An additional consideration is that FGFR3 expression was found to be higher in normal brain tissue compared to GBM (Fig. 1). Since pemigatinib also targets FGFR3, potential effects on healthy tissue must be taken into account. Two factors may mitigate this concern. First, although pemigatinib is a potent inhibitor of FGFR1–3, it displays markedly reduced activity against the majority of non-FGFR kinases (>100-fold selectivity), thereby limiting broad off-target kinase inhibition. Second, while the blood–brain barrier (BBB) is frequently disrupted within GBM lesions, allowing drug penetration into tumor areas, it remains largely intact in surrounding non-tumoral regions. This differential permeability is likely to restrict drug access to FGFR3-expressing normal parenchyma. Consistent with this, clinical data from trials such as FIGHT-202 have reported primarily on-target adverse effects, including hyperphosphatemia due to altered phosphate handling, along with mucosal toxicities, nail and skin changes, and ocular events. These effects are manageable and largely reversible with supportive measures such as phosphate binders, local care, and dose adjustments [33,34]. Neurological adverse events have not been a prominent feature, suggesting limited CNS toxicity.
Another important issue is the possibility of acquired resistance to pemigatinib. In other cancers such as cholangiocarcinoma, resistance has been attributed to secondary mutations in the FGFR kinase domain or activation of alternative signaling pathways. Although such mechanisms have not yet been described in GBM, the intrinsic plasticity of glioblastoma stem cells and the frequent activation of bypass pathways (e.g., EGFR, MET, PDGFR) suggest that resistance could also emerge in this context. In light of our findings showing context-dependent radiosensitization, further studies will be needed to determine whether combining pemigatinib with radiotherapy or temozolomide could help prevent or delay resistance in a subset of GBM. Clinical evaluation will ultimately be required to clarify the efficacy and feasibility of such strategies”.
Minor observations
Comments 22. There are multiple typos, including critical ones (e.g., "Mice received pemigatinib (0. mg/kg, oral gavage)" or incorrect use of FoxM1 instead of FOXM1)
Response 22. We thank the reviewer for pointing out typographical and nomenclature issues. We have carefully proofread the entire manuscript and corrected all typographical and nomenclature issues across text, figures, legends, and SI: 0.5 mg/kg (not “0. mg/kg”); standardized HGNC capitalization (FOXM1, FGFR1–3, S100A4, HIF-1α); fixed unit spacing. We believe these corrections address the reviewer’s concerns.
Comments 23. Some figures show digital editing in the embedded text.
Response 23. Thank you for pointing this out. We performed a full audit of all main and supplementary figures and corrected every instance of misaligned or inconsistently formatted text.
Comments 24. Avoid non-quantitative terms such as "lower doses" or "clear radiosensitization". Provide numerical values and refer to the corresponding results or figures.
Response 24. Thank you for the comment. We have corrected this and ensured that non-quantitative terms are limited and accompanied by exact values.
Comments 25. Use SI units and abbreviations consistently throughout the manuscript.
Response 25. Thank you for the suggestion. We performed a manuscript-wide audit and made the corrections.
Reviewer 3 Report
Comments and Suggestions for Authors
I deeply appreciate the opportunity to review your fascinating manuscript on FGFR biology. After careful examination, I would like to offer the following suggestions for your consideration:
The methodological descriptions in sections 2.9 (RNA sequencing), 2.10 (fusion and mutation determination), and 2.11 (bioinformatics analysis) require substantial expansion. Currently, these critical procedures lack sufficient technical detail—please elaborate on sample handling protocols, sequencing methodology, and validation approaches to strengthen reproducibility.
Several figures contain misaligned text elements that detract from their clarity. These formatting issues warrant attention before submission.
The manuscript would benefit from cytotoxicity studies using non-transformed cells to contextualize your findings within a therapeutic index framework.
While your use of two cell lines with differential methylation patterns provides valuable insights, incorporating additional discussion around EGFR and IDH status would enrich your narrative. Furthermore, data from complementary cell lines, if available, would broaden the applicability of your conclusions.
Have you explored potential effects on other FGFR family members (particularly FGFR2, FGFR3) or cross-talk with EGFR signaling? Such data would significantly enhance mechanistic understanding.
For the si100A4 knockdown experiments, representative immunoblots demonstrating efficient protein reduction would strengthen the validity of your functional observations.
I noticed Figure 5D lacks error bars for the GC2 experimental condition—this statistical information should be included.
Regarding your in vivo work, toxicity parameters such as body weight trajectories would provide valuable safety context for the observed antitumor effects.
Author Response
Comment 1: The methodological descriptions in sections 2.9 (RNA sequencing), 2.10 (fusion and mutation determination), and 2.11 (bioinformatics analysis) require substantial expansion. Currently, these critical procedures lack sufficient technical detail—please elaborate on sample handling protocols, sequencing methodology, and validation approaches to strengthen reproducibility.
Response 1: We thank the reviewer for this important request. We have substantially expanded Sections 2.9 (RNA sequencing), 2.10 (Fusion and mutation determination), and 2.11 (Bioinformatics analysis) to include detailed information.
Material and Methods (Section 2.9, page 5, lane 205)
“2.9 RNA sequencing
For transcriptomic analysis, GC1 and GC2 glioblastoma stem cell lines were seeded at 2 × 10⁶ cells in 5 mL medium in 25 cm³ flasks and treated with either 250 nM pemigatinib or DMSO control for 48 h (three independent biological replicates per condition per cell line, n = 6 total per condition). RNA was extracted using the RNeasy kit (Qiagen) and quality verified (RQN > 8) prior to library preparation with the Illumina Stranded Total RNA Prep protocol, including rRNA depletion, strand-specific library construction, and paired-end sequencing. Library quality and concentration were assessed using Qubit™ dsDNA BR Assay (Thermo Fisher Scientific), High Sensitivity NGS Fragment Analysis Kit (Agilent Technologies), and KAPA Library Quantification Kit for Illumina® (Roche). Sequencing was performed on an Illumina NextSeq 550 with a High Output flow cell, and raw reads were demultiplexed with bcl2fastq v2.20.0.422. Transcriptomic results shown in Figure 4 represent an integrated differential expression analysis across both models, averaged to highlight changes conserved between GSC lines.
2.10 Fusion and mutation determination
Reference and alignment. Reads were aligned to GRCh38 using STAR v2.7.x in two-pass mode with default gene model guidance (GTF: Ensembl release v98). Soft clipping and chimeric junction output were enabled for fusion discovery.
Fusion calling. Gene fusions were called with Arriba v2.x (using the recommended blacklist and database bundles [21] and STAR-Fusion v1.x (with the CTAT resource library built on GRCh38; Ensembl v98) [22]. Minimal evidence requirements followed tool defaults (e.g., Arriba: ≥1 junction read; STAR-Fusion: junction + spanning support). Results were filtered to remove read-through events, paralog artifacts, mitochondrial fusions, and panel-of-normal hits. High-confidence fusions were defined as (i) detected by both callers or (ii) single-caller calls with strong supporting evidence and manual confirmation in IGV. Under these criteria, no high-confidence FGFR1/2/3 fusions were detected in GC1 or GC2.
Variant calling from RNA-seq. Putative variants in FGFR1–3 were identified from RNA-seq using GATK HaplotypeCaller (gatk-4.2.0.0, GRCh38v98,https://gatk.broadinstitute.org/hc/en-us/articles/360037225632-HaplotypeCaller) following RNA-seq best practices: SplitNCigarReads, base quality recalibration, and variant calling restricted to coding regions of canonical transcripts (Ensembl v98). Variants were filtered with hard thresholds (e.g., low depth/quality or strand bias) and annotated by VEP (SIFT/PolyPhen) with cross-reference to ClinVar, COSMIC, CIViC. Variants were categorized as pathogenic/likely pathogenic, benign/likely benign, or VUS. Consistent with the Results, only VUS (and known SNPs) were found in FGFR1–3; no activating kinase-domain mutations were identified. Selected loci were reviewed visually in IGV to confirm read support and rule out mapping artifacts. This analysis was conducted using Alamut software.
2.11 Bioinformatics analysis
QC and pre-processing. FASTQ quality was assessed with FastQC and summarized with MultiQC (per-base quality, adaptor content, GC distribution). Post-alignment QC included mapping rate, multimapping, rRNA content, gene body coverage, strandedness, and duplication estimates (Picard).
Quantification and DE. Gene-level counts were generated with featureCounts (union exon model; stranded = reverse for the Illumina stranded protocol). Differential expression was performed in R/DESeq2 with the design ~ cell_line + treatment to estimate the treatment effect while accounting for GC1/GC2 differences. Multiple testing was controlled by Benjamini–Hochberg; unless noted, genes with adjusted p < 0.05 and |log₂FC| > 1 were considered significant. For Fig. 4A, log₂FC values represent line-averaged effects when regulation was concordant across GC1 and GC2.
Pathway and enrichment analyses. The mRNA expression levels of FGFR receptors and S100A4 in GBM tissue compared to normal tissue were evaluated using Gene Expression Profiling Interactive Analysis (GEPIA, http://gepia.cancer-pku.cn/), a web-based tool that provides fast and customizable analyses based on TCGA and GTEx data. For this analysis, the log2 fold change (log2FC) cutoff was set at 1, and the p-value cutoff was set at 0.01. Gene expression values are presented as log₂(TPM + 1), where TPM denotes transcripts per million.
A volcano plot was generated to visualize the differential gene expression in GSCs treated with or without pemigatinib, using RNA-seq data and the Srplot tool (Scientific and Research plot tool, http://www.bioinformatics.com.cn/SRplot). This free online platform was also used to visualize the principal pathways affected in GSCs treated with pemigatinib. Impacted and downregulated genes were identified through Gene Ontolomg (GO) analysis using the web-based portal Metascape (https://metascape.org).”
Comment 2: Several figures contain misaligned text elements that detract from their clarity. These formatting issues warrant attention before submission.
Response 2: Thank you for flagging this. We performed a figure-wide formatting audit and corrected all instances of misalignment.
Comment 3: The manuscript would benefit from cytotoxicity studies using non-transformed cells to contextualize your findings within a therapeutic index framework.
Response 3: We thank the reviewer for this important suggestion. Our study evaluated both established glioblastoma lines (LN18, U87) and patient-derived glioblastoma stem cells (GC1, GC2), the latter being primary tumor cells that model the therapy-resistant compartment. We agree, however, that including non-malignant CNS cells would better define a therapeutic window.
Although such experiments were beyond the scope of the present work, clinical experience with pemigatinib (e.g., FIGHT-202 and subsequent studies) indicates a predominantly on-target, manageable safety profile—hyperphosphatemia related to phosphate handling, mucosal toxicities, nail/skin changes, and ocular events—without prominent neurological adverse effects at therapeutic doses (Abou-Alfa, Lancet Oncol 2020; Vogel, ESMO Open 2024). While clinical safety data are not a substitute for normal-cell in vitro testing, they support overall tolerability and the absence of obvious CNS toxicity.
Nevertheless, we acknowledge the value of including direct comparisons with non-transformed CNS cells in preclinical settings to strengthen the case for a favorable therapeutic window. We have now included this point in the Discussion section of the revised manuscript and propose to explore this in future studies using human astrocytes or neural progenitor models.
Changes in the manuscript:
Discussion (Section 4, page 20, lane 749)
" Additionally, we did not include non-malignant CNS cells (e.g., human astrocytes or neural progenitor/neuronal models), which would allow estimation of a therapeutic index. Clinical experience with pemigatinib indicates predominantly on-target, manageable toxicities without prominent neurological adverse events [36,37], but head-to-head assays in normal CNS models will be important to define selectivity. "
Comment 5: Have you explored potential effects on other FGFR family members (particularly FGFR2, FGFR3) or cross-talk with EGFR signaling? Such data would significantly enhance mechanistic understanding.
Response 5: We thank the reviewer for this suggestion. Our mechanistic focus was on FGFR1 because prior work from our group (Deshors, 2025) showed at the protein level that in GC1/GC2 FGFR1 is strongly expressed, FGFR2 is undetectable, and FGFR3/FGFR4 are present at much lower levels relative to FGFR1. In the present study we did not perform dedicated activity assays for FGFR2/3/4. We have revised the Discussion to (i) clarify this rationale based on Western blot evidence, (ii) add literature supporting EGFR–FGFR cross-talk : independent studies have demonstrated feedback activation of EGFR upon FGFR inhibition and improved activity with dual FGFR–EGFR blockade (Wu et al., Cancer Discovery, 2022). Additional work has revealed bidirectional trafficking/signaling interdependence between FGFR and EGFR at clathrin-coated sites, suggesting receptor cross-regulation at the plasma membrane (Alfonzo-Méndez et al., bioRxiv, 2024). More broadly, EGFR and FGFR converge on RAS/MAPK, PI3K/AKT, and STAT pathways; EGFR upregulation can act as a bypass upon FGFR inhibition—mechanisms that may be particularly pertinent in GBM, where EGFR alterations are frequent.
Discussion (Section 4, page 19, lane 686)
« In addition to biomarker-driven selection, receptor cross talk may also shape response to FGFR inhibition. Although pemigatinib inhibits FGFR1–3, we centered our analyses on FGFR1 because protein-level profiling in patient-derived GC1/GC2 [33] showed robust FGFR1, no detectable FGFR2, and comparatively low FGFR3/FGFR4. Given the extensive RTK cross-talk in GBM, EGFR amplification/variants (e.g., EGFRvIII) may provide bypass signaling that limits the impact of FGFR1 blockade. Consistent with this, feedback activation of EGFR upon FGFR inhibition and improved activity with dual FGFR–EGFR block-ade have been reported [34], and receptor co-regulation at clathrin-coated sites supports direct EGFR–FGFR interplay at the plasma membrane [35].
Comment 6: For the si100A4 knockdown experiments, representative immunoblots demonstrating efficient protein reduction would strengthen the validity of your functional observations.
Response 6: We thank the reviewer for this valuable suggestion. In response, we evaluated S100A4 protein expression following siRNA-mediated silencing in GC1 and GC2 glioblastoma stem cells by flow cytometry. The analysis showed a reduction of 37–49% in GC1 and 55% in GC2 after transfection with two independent siRNAs compared with the control siRNA (siCtrl), consistent with our mRNA expression results. These data have been added to the revised manuscript (new Figure 5D).
RESULTS (Section 3.8, page 14, lane 527)
“Flow cytometry analysis confirmed efficient S100A4 silencing at the protein level, showing a 37-49% reduction in GC1 and a 55% reduction in GC2 compared with siCtrl, in line with the mRNA expression data (Fig 5D).”
Fig 5D (Page 15, lane 537)
Insertion Fig5D
Legend Fig 5D (page 16, lane 553)
“(D) S100A4 protein levels in GC1 and GC2 cells after transfection with control siRNA (siCtrl) or two independent siRNAs targeting S100A4 (siS100A4(1) and siS100A4(8)) were analyzed using flow cytometry.”
Comment 7: I noticed Figure 5D lacks error bars for the GC2 experimental condition—this statistical information should be included.
Response 7: Thank you for pointing this out. You are correct—error bars for the GC2 condition were missing in the original figure. In the revised manuscript, this panel is now Fig. 5E; we have added error bars (mean ± SD) for all conditions.
Comment 8: Regarding your in vivo work, toxicity parameters such as body-weight trajectories would provide valuable safety context for the observed antitumor effects.
Response 8: We agree and have now included body-weight trajectories for all groups (CTL, IR, Pem, Pem+IR) as Supplementary Fig. S3. The figure reports percent change from baseline (week 0) to week 3, mean ± SD, with SDs derived from the SDs of absolute weights by error-propagation. Over this interval, trajectories were similar across arms, and we did not observe an early weight-loss signal with pemigatinib (0.5 mg/kg, PO, 5 days/week) or with the combination.
Results (section 3.9, page 16, lane 565)
“Body weight was monitored weekly for the first 3–4 weeks. Trajectories (% change from baseline; mean ± SD) were comparable between groups, without an early weight-loss signal in the pemigatinib or combination arms (Supplementary Fig. S3).”
Figure S3 (Appendix D, page 25, lane 868)
Insertion Fig S3
Legend Fig S3 (page 25, lane 871)
“Figure S3. Body-weight trajectories (% change from baseline). Mean ± SD of percent change in body weight from baseline (week 0) to week 3 for the CTL, IR, Pem, and Pem+IR cohorts. Error bars represent SD; at each time point, SDs were derived from the SDs of absolute body weights using error propagation.”
Round 2
Reviewer 1 Report
Comments and Suggestions for Authors
In this version of the article, our comments were taken into account, and the statement on the practical application of the data obtained was softened. Thus, the conclusions have become more correct and further corrections are not required.
Author Response
Comment 1. In this version of the article, our comments were taken into account, and the statement on the practical application of the data obtained was softened. Thus, the conclusions have become more correct and further corrections are not required.
Response 1. We thank the reviewer for this positive assessment and are pleased that the revisions have addressed the concerns raised. We appreciate the constructive feedback, which has helped us improve the clarity and accuracy of our conclusions.
Reviewer 3 Report
Comments and Suggestions for Authors
Thank you for the opportunity to review the revised article. I appreciate it. I feel the toxicity test would strengthen your manuscript.
Author Response
Comment 1: Thank you for the opportunity to review the revised article. I appreciate it. I feel the toxicity test would strengthen your manuscript.
Response 1: We thank the reviewer for this constructive suggestion. We agree that a toxicity assessment would further strengthen the manuscript. However, we did not include such experiments in the present study since the molecule investigated is already used in the clinic, where its safety profile has been established.